



# A hydrography upscaling method for scale invariant parametrization of distributed hydrological models

Dirk Eilander[1,2], Willem van Verseveld[2], Dai Yamazaki[3], Albrecht Weerts[2,4], Hessel C. Winsemius[2,5], Philip J. Ward[1]

[1]Institute for Environmental Studies (IVM), Vrije Universiteit Amsterdam, Amsterdam, The Netherlands
[2]Deltares, Delft, The Netherlands
[3]Institute of Industrial Sciences, the University of Tokyo, Tokyo, Japan
[4]Hydrology and Quantitative Water Management Group, Wageningen University & Research, Wageningen, The Netherlands
[5]Dar Es Salaam Resilience Academy, Dar Es Salaam, Tanzania

*Correspondence to*: Dirk Eiland (dirk.eilander@deltares.nl)

**Abstract.** Distributed hydrological models rely on hydrography data such as flow direction, river length, slope and width. For large-scale applications, many of these models still rely on a few flow-direction datasets, which are often manually derived. We propose the Iterative Hydrography Upscaling (IHU) method to upscale high-resolution flow direction data to the typically coarser resolutions of distributed hydrological models. The IHU aims to preserve the upstream-downstream relationship of river structure, including basin boundaries, river meanders and confluences, in the D8 format, which is
commonly used to describe river networks in models. Additionally, it derives sub-grid river attributes such as drainage area, river length, slope and width. We derived the multi-resolution MERIT Hydro IHU dataset at resolutions of 30 arcsec (~1km), 5 arcmin (~10 km) and 15 arcmin (~30 km) by applying IHU to the recently published 3 arcsec MERIT Hydro data. Results indicate improved accuracy of IHU at all resolutions studied compared to other often applied methods. Furthermore, we show that using IHU-derived hydrography data minimizes the errors made in timing and magnitude of simulated peak
discharge throughout the Rhine basin compared to simulations at the native data resolutions. As the method is fully automated, it can be applied to other high-resolution hydrography datasets to increase the accuracy and enhance the uptake of new datasets in distributed hydrological models in the future.

## 1 Introduction

Large-scale distributed hydrological and land surface models are used to provide estimates of available water resources
(Schewe et al., 2014; Wada et al., 2011), flood risk (Hirabayashi et al., 2013; Ward et al., 2013), drought risk (Veldkamp et al., 2017) and food production (Kummu et al., 2014), amongst other applications. These models generally contain a routing module to simulate streamflow, i.e.: the lateral flow of water on the land surface. This is a key variable for understanding the water, energy and biogeochemical cycles and the effects of disturbances from anthropogenic climate change on these cycles (Wood et al., 2011). The spatial pattern of average streamflow conditions is largely determined by the contributing area of a
river segment (Quinn et al., 1991), which is imposed on a model by its flow direction data. Simulated peak streamflow is





particularly sensitive to the accuracy of the flow directions and river channel properties (Paiva et al., 2013) and very important at river confluences (Geertsema et al., 2018; Guse et al., 2020; Metin et al., 2020) and river outlets (Couasnon et al., 2020; Eilander et al., 2020), where multiple fluvial and/or coastal flood drivers may combine to modulate a flood event. Furthermore, streamflow is the only measurable integral signal of basin response and is therefore widely used for model

calibration (Bouaziz et al., 2020), underlining the importance of flow direction data in distributed hydrological models.

Over the last decade, large-scale distributed hydrological models have been applied at increasingly higher resolutions (Bierkens, 2015), which poses a challenge on the parametrization of these models (Wood et al., 2011). The Multiscale Parameter Regionalization method (Kumar et al., 2013; Samaniego et al., 2010), which relates coarse-resolution model parameters to physiographic high-resolution data through different transformation and upscaling techniques, was proposed as

a method to obtain model parameters that can be transferred across spatial scales. This method does not, however, consider hydrography data, which has been recognized as a limitation (Thober et al., 2019) as it is a source of difference in discharge prediction skill between resolutions (Imhoff et al., 2020). Scale-invariant parametrization of hydrological models requires consistent flow direction as well as sub-grid river channel parameters such as length and slope across resolutions.

Flow direction data in distributed hydrological models is commonly described in the "deterministic eight neighbors" (D8)

format, which sets the downstream direction of each cell to one of its eight neighboring cells. Flow directions are typically derived from elevation data based on the direction with the steepest slope (e.g. Lehner et al., 2008). Flow directions and river slope and length parameters cannot be accurately inferred from coarse resolution elevation data as this data is no longer representative for the elevation of streams. Instead, the flow directions should be derived by upscaling finer resolution flow direction data (e.g. Yamazaki et al., 2009). At the same time new high-resolution global hydrography datasets, such as the 3

arcsec MERIT Hydro data (Yamazaki et al., 2019), are becoming available at spatial resolutions that are finer than current state-of-the-art large-scale distributed hydrological models (typically ≥ 1 km). Therefore, automated upscaling methods are required to describe high resolution flow directions and river parameters at coarser resolutions to leverage these new datasets for distributed hydrological modelling.

Several D8 flow direction upscaling methods have been developed, including: the Network Scaling Algorithm (NSA; Fekete

et al., 2001); the Double Maximum Method (DMM; Olivera et al., 2002); the Effective Area Method (EAM; Yamazaki et al., 2008); and the hierarchical dominant river tracing (DRT) algorithm (DRT; Wu et al., 2011). Most of these methods first determine which river to represent within each cell and subsequently set the upscaled flow direction based fine-resolution flow directions. However, to correctly determine which river to represent within a cell in order to preserve the river network often requires more information than contained in just one cell. Therefore many commonly used coarse-resolution flow

direction datasets, such as DDM30, were initialized based on an automatic upscaling method, such as DMM, but required manual corrections to ensure the river network is well preserved (Döll and Lehner, 2002). To solve this issue, the more recently developed DRT uses global information to set flow directions for large rivers using a hierarchical approach. To automatically preserve large rivers at coarser resolutions DTR reroutes rivers through neighboring cells if required. This method has proven successful at automatically upscaling 30 arcsec flow direction data to coarser resolutions up to 30 arcmin





(Wu et al., 2012), but its application is limited at higher resolutions as it requires global information, thus entire basins to be loaded in memory before processing. Furthermore, none of these upscaling methods derive sub-grid river parameters such as length and slope, which are required for scale-invariant parametrization of hydrological models.

The first objective of this paper is therefore to develop a fully automated flow direction upscaling algorithm in order to derive scale invariant flow direction and sub-grid drainage area, river length and slope parameters that can be applied to

high-resolution (< 1 km) global hydrography data. The second objective is to evaluate how the choice of upscaling method and resolution affect peak discharge simulation. The paper is set up as follows. Section 2 describes the new upscaling method, the method to derive sub-grid river variables, the baseline and output datasets, and the benchmark and case study experiments. Section 3 presents the results of the benchmark of IHU against DMM and EAM at the global scale. Section 4 presents the results of a case-study in which we test the scale-invariance of simulated peak discharge. The results are

discussed in section 5 and conclusions based on this study are presented in section 6.

## 2 Method

The flow direction upscaling method proposed in this paper, the Iterative Hydrography Upscaling (IHU) method, is described in section 2.1 and implemented in the open source python *pyflwdir v0.4.4* package (http://deltares.gitlab.io/wflow/pyflwdir). Section 2.2 describes the method to derive sub-grid parameters of drainage area,

river length and slope. Section 2.3 discusses the upscaled *MERIT Hydro IHU v1* dataset, which is released as part of this paper. Section 2.4 describes the metrics used to evaluate the accuracy of the upscaling methods. Section 2.5 describes a case study for the Rhine basin used to assess the effect of upscaling method and resolution on simulated discharge.

### 2.1 The Iterative Hydrography Upscaling algorithm

The IHU is explained in this section and illustrated for a fictional river in Figure 1, where the used terminology is explained

in the legend. IHU requires a target coarse-resolution grid definition (grey dashed lines), often defined by a multiple of the fine-resolution grid, and two input maps: a fine-resolution flow direction and upstream area map (blue lines, where darker blue indicates a larger river). For convenience, we refer to target coarse-resolution target raster cells as *cells* and fine-resolution raster cells as *pixels*. The goal of the upscaling method is to define the most representative flow direction for each cell (arrows).

The IHU method exists of four iterations which all consist of three steps. It builds on the Effective Area Method (EAM; Yamazaki et al., 2008) as it shares the same initialization, see Appendix A for a detailed description of EAM. The iterations are numbered and aimed at: (row 1) initiating flow directions; (row 2) fixing erroneous flow directions; (row 3) optimizing the in-between outlet pixel distance; and (row 4) minimizing the error made when a cell cannot be connected. The steps of each iteration are lettered: (column A) initiate, (column B) analyze and (column C) update. Each step is explained in detail

below and refers to a panel of Figure 1. Iteration 2-4 can be repeated to improve the results.

**Figure 1: Illustration of the Iterative Hydrography Upscaling (IHU) method. Firstly, 1A) for each cell the representative river pixel (dark red square) inside the effective area (shaded area) and subsequently the outlet pixel (orange square) is identified, and 1B) based on the fine-resolution flow path downstream of the outlet pixel (black lines), 1C) initial flow directions (orange arrows) are set. Secondly, 2A) erroneous flow-directions (red arrows) are identified, and 2B) analyzed in context of the fine-resolution downstream flow-path (black line) with alternative outlet pixels (green squares) and tributary outlet pixels (grey squares), to 2C) fix the flow directions by relocating outlet pixels (orange square and arrows). Thirdly, 3A) outlet pixels with short in-between**





distance are identified (red squares), and 3B) alternative outlet pixels (green squares) with sufficient in-between distance are searched, after which 3C) outlet pixels are relocated and flow directions are updated accordingly (orange square and arrows).
Fourthly, 4A) remaining erroneous flow directions are identified (red arrows), and 4B) from each neighboring cell the distance to a common downstream outlet pixel (green square) is measured, to 4C) update the flow directions (orange arrow) to the neighboring cell with the minimum distance in order to minimize upscaling errors.

**Step 1A:** The first iteration sets an initial flow direction for each cell. First, a representative river pixel is found for each cell (dark red square). This pixel is defined as the river pixel with the largest upstream area within the effective area (grey shade),
as described by equation 1. Then, that pixel is traced downstream towards the outlet pixel (orange square), which is set as the most downstream pixel before leaving the cell. This first step of IHU builds on EAM as it uses the same starting point to identify an initial representative river pixel per cell. By defining the effective area for selecting the representative river pixel in each cell, the method avoids selecting river segments that only pass through a corner of a cell and are unfavorable to determine flow directions (Paz et al., 2006).

$$\{(x,y)|((x - x_0)^{0.5} + (y - y_0)^{0.5}) < R^{0.5}\}\,, \tag{1}$$

where, $x, y$ are the coordinates of a pixel, $x_0, y_0$ are the coordinates of the center of a cell and R is half the cell size.

**Step 1B**: The outlet pixel of each cell (grey square) is traced downstream (black lines) until an outlet pixel in a neighboring downstream cell is found. If no outlet pixel is found before leaving the eight neighboring cells, the trace is ended at the first pixel inside the effective area downstream of the outlet pixel, which is the default EAM procedure.

**Step 1C**: The initial upscaled flow direction (orange arrows) is set for each cell in the direction of the cell where the trace in step 1B ends.

**Step 2A**: The second IHU iteration aims to conserve fine-resolution flow directions between outlet pixels at the coarse resolution, by repairing erroneous flow directions. The flow direction of a cell is erroneous if the first outlet pixel downstream of the outlet pixel of that cell is not located in its downstream cell (i.e. where the flow direction points to). In
this step erroneous flow directions (red arrows) are identified. In the example erroneous flow directions are identified in *cell h and x*, as the outlet pixels downstream of these cells are not located in their downstream cells (i.e.: to the east for *cell h* and to the northwest for *cell x*). Step 2B and 2C are then executed for each cell with erroneous flow direction, sorted from cells with a small to large upstream area at the outlet pixel, and iterated until no more flow direction can be corrected.

**Step 2B** (illustrated for *cell h only*): The outlet pixel of a cell with erroneous flow direction (black square) is traced
downstream (black line) while potential alternative outlet pixels (green squares) are identified: these are defined as the last pixel before entering a new cell on the trace. The trace ends at the next downstream outlet pixel of a cell with correct flow direction and with only one potential outlet pixel. Cells directly upstream of cells with (alternative) outlet pixels on the trace are marked as tributary cells and their outlet pixels as tributary outlet pixels (grey squares). For all tributary cells the first and last alternative outlet pixel to which a valid flow direction can be set are identified. The erroneous flow directions are then
updated based on the following iterative procedure:





Starting from the most upstream outlet pixel on the trace, an outlet pixel is relocated to the most downstream alternative outlet pixel in a neighboring cell for which flow directions from the upstream and all tributary outlet pixels can be set correctly. If required to set the flow directions of tributary cells correctly, an alternative outlet pixel can be set in headwater cells (i.e. cells without upstream neighbors) to connect the tributary cell to a downstream outlet pixel. This is repeated until

the end of the trace is reached.

If at some point no valid alternative outlet pixel is found, the position of the last relocated outlet pixel is flagged as a bottleneck and not considered in the next iteration. Note that there are no bottlenecks in the example.

This step is iterated until successful or no new bottlenecks are found.

**Step 2C**: If step 2B is successful the flow directions are updated accordingly. In this example the outlet pixel of *cell o* is

relocated (from black to orange square), thereby changing the flow direction for *cells h and u* (orange arrows). The first outlet pixel downstream of *cell h* is now located in its southeast neighboring cell where the outlet pixel is relocated to the main stream. The first outlet pixel downstream of *cell u* is located in its northeast neighboring cell. Note that the flow direction of *cell x* cannot be repaired because two rivers flow parallel in its downstream *cell q* of which only one can be represented at the output resolution.

**Step 3A**: The third iteration aims to optimize the distance in-between outlet pixels, measured along the fine-resolution flow directions. If this distance is short, one of the outlet pixels can potentially be removed in favor of another river segment within the same cell. A short in-between outlet pixel distance is not favorable when this distance is used to set the river segment length in routing models as it will decrease the accuracy or require smaller timesteps. In this step, outlet pixels with an in-between outlet pixel distance smaller than a threshold value are flagged (red squares). This threshold is set to 25% of

the length of a cell edge resulting in flagged outlet pixels for *cell m and n* in the example. Then, steps 3B and 3C are executed for every cell with a flagged outlet pixel until no more outlet pixels are relocated.

**Step 3B**: First, it is checked whether a flagged outlet pixel can be removed while the flow directions of its upstream neighboring cells can be set correctly. Then, alternative outlet pixels within the same cell are identified (green square). Alternative outlet pixels should have a minimum upstream area; a minimum distance to the next outlet pixel and may not be

located downstream of any other outlet pixel. The minimum upstream area threshold is set to 25% of the cell area. In the example an alternative outlet pixel is found in *cell m*.

**Step 3C**: If one or more alternative outlet pixels are found for a cell in step 3B, the outlet pixel is relocated to the alternative outlet pixel with the largest upstream area, and the flow directions are updated accordingly. In the example the outlet pixel of *cell m* is relocated (orange square), thereby changing the flow direction for *cells m and n* (orange arrows). The first outlet

pixel downstream of *cell m* is now located in the cell to the east because the *cell m* now represents another stream. The first outlet pixel downstream of *cell n* is now located in the cell to the north as the original outlet pixel in *cell m* is relocated.

**Step 4A**: This iteration aims to minimize upscaling errors where erroneous flow directions cannot be repaired. This occurs mostly where multiple rivers flow parallel through the same cell while one can be represented in the D8 format. First, cells with remaining erroneous flow directions after step 2 are identified (red arrows). Then, step 4B and 4C are executed for each



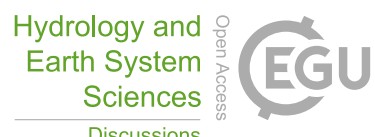

identified cell, sorted from cells with a large to small upstream area at the outlet pixel. In the example, the flow direction of *cell x* is erroneous as two rivers flow parallel through its downstream *cell q*.

**Step 4B**: The fine-resolution path downstream of a cell with erroneous flow direction is traced (black line) and outlet pixels on the trace are identified (green squares). For each neighboring cell the coarse-resolution flow-direction is followed until it reaches an outlet pixel on the trace or a maximum length of 100 cells. The distance to this outlet pixel from the neighboring

cell and to this outlet pixel from the cell with erroneous flow directions are measured in number of outlet pixels and summed. This yields a combined distance to a common downstream outlet pixel for each neighboring cell. If multiple neighboring cells have the same combined distance, the cell with the largest upstream are at the outlet pixel is selected as downstream pixel. Finally, if setting the flow direction to a neighboring cell yields the flow directions from two adjacent cell to cross, this cell is not considered. In the example, combined distance from the pixel of *cell x* and neighboring cells *cell q, r*

*and w* to a common downstream outlet pixel are calculated.

If no downstream outlet pixel is found on the trace and the last pixel of the trace is at a river mouth or sink, that pixel is set as outlet pixel in the cell with erroneous flow direction if within 2 cells distance. Note that in this case the outlet pixel is located outside the cell where it belongs to. If iteration 2-4 are repeated, this step is only executed in the last repeat. This situation occurs if a larger river flows through the cell with the river mouth or sink pixel. This step preserves secondary

rivers in cells with larger rivers or multiple river outlets or sinks.

**Step 4C**: The flow direction is updated (orange arrows) to the neighboring cell with the shortest combined distance to a common downstream outlet pixel (green square). In the example the shortest combined distance from *cell x* is found in *cell r* to the common downstream outlet pixel in *cell k*, changing the flow direction of *cell x* to north. While this introduces a small error in *cell r*, the error is contained to just that cell minimizing the upscaling error.

**2.2 Sub-grid hydrography variables**

Based on fine-resolution flow direction and outlet pixels as derived from IHU, see previous section, several sub-grid variables are derived as shown in Figure 2:

- The *sub-grid area* is defined by the pixels draining to the outlet pixel of a cell and is confined by upstream outlet pixels, see Figure 2B. This area is also referred to as the unit-catchment area as introduced by Yamazaki et al

195      (2009).

- The *river length* is defined by the fine-resolution flow path found by tracing the outlet pixel of a cell either up- or downstream until the next outlet pixel, see Figure 2C-D. When tracing a pixel upstream, the upstream pixel with the largest upstream area is selected in case of multiple upstream pixels. The length is calculated along the sub-grid flow path based on the pixel size, with diagonal steps are taken to be $\sqrt{2}$ times the pixel size. Both up- and

downstream river lengths are used in routing models and derived here.





- The *river slope* is based on the MERIT Hydro elevation difference between two pixels at a set distance up- and downstream of the outlet pixel. Here we used a default distance of 2 km, 1 km up- and downstream of the outlet pixel. The flow path along which the slope is derived is shown in Figure 2E.

- The *river width* is based on the MERIT Hydro width at the outlet pixel. Note that this data contains gaps, i.e. not all outlet pixels have a river width in the underlaying data, see Figure 2F, which need to be filled to achieve global coverage of river width. Filling of the river width data gaps was only done for a case study, see section 2.5.

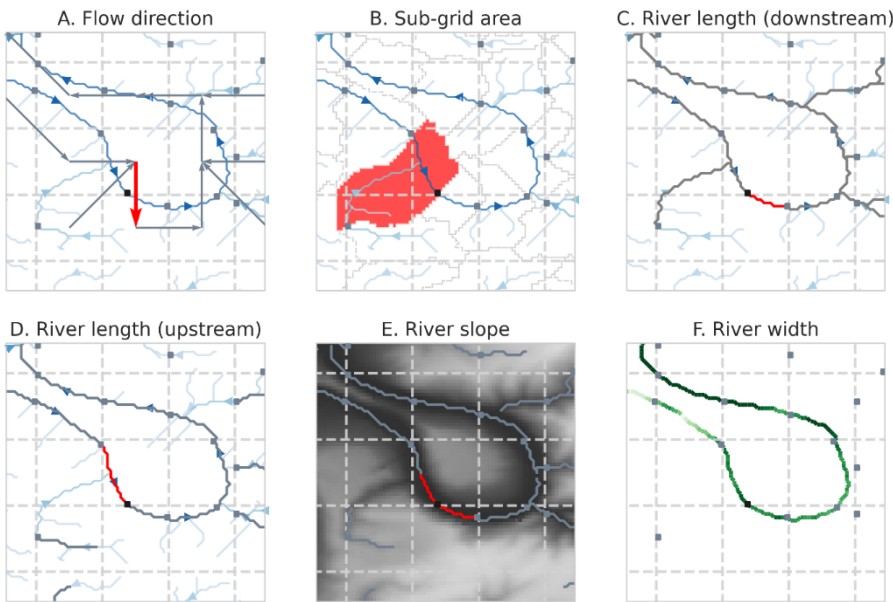

**Figure 2: Output hydrography variables based on fine-resolution flow directions (blue arrows in A-D; darker indicates larger**
**upstream area) and/or outlet pixels (squares). Each variable is highlighted in red for the center cell and grey for other cells. The sub-grid area (B) is based on all pixels draining the outlet pixel and limited by upstream outlet pixels. River length is derived based on the length of the flow path from outlet pixel to next downstream (C) or upstream (D) outlet pixel. The River slope (E) is calculated as the elevation (grey colors) difference over a flow path from a set length upstream to downstream of the outlet pixel. The river width (F) is derived based on the sub-grid river width (green colors) at the outlet pixel location.**

**2.3 Multi-resolution hydrography dataset**

We derived the multi-resolution MERIT Hydro IHU dataset at resolutions of 30 arcsec (~1km), 5 arcmin (~10 km) and 15 arcmin (~30 km) by applying IHU to the recently published 3 arcsec MERIT Hydro data (Yamazaki et al., 2019). The original MERIT Hydro data were near-automatically derived based on the MERIT DEM (Yamazaki et al., 2017) and several water body datasets and show good agreement with drainage areas reported by the Global Runoff Data Center (GRDC). We
selected this MERIT Hydro as it has a larger spatial coverage (N90 to S60) and better representation of small streams (Yamazaki et al., 2019) compared to earlier published hydrography datasets. It also provides supplementary data layers including hydrologically adjusted elevation, which is used to derive sub-grid river slope, and river channel width derived





from the G1WBM permanent water body layer (Yamazaki et al., 2014), which is used to derive sub-grid river width. An overview of the layers in the upscaled *MERIT Hydro IHU* dataset is given in Table 1.


**Table 1: Overview of hydrography and metadata layers of the MERIT Hydro IHU v1 dataset.**

| Parameter | Name | Unit | Description |
|---|---|---|---|
| **Hydrography** | | | |
| Flow direction | flwdir | - | D8 flow directions |
| River length | rivlen | m | Sub-grid distance between two outlet pixels along the flow path, diagonal steps are taken to be $\sqrt{2}$ times the pixel size. River length in the downstream direction has a " ds" postfix |
| River slope | rivslp | $\text{mm}^{-1}$ | Average slope based on the elevation difference between pixels at a set distance of 2 km around (1 km up- and downstream) the outlet pixel |
| River width | rivwth | m | Width at sub-grid outlet pixel |
| Sub-grid area | subare | $\text{m}^2$ | Sum of sub-grid pixel areas draining to the pixel outlet confined by the upstream sub-grid pixel(s) |
| Upstream area | uparea | $\text{km}^2$ | Accumulated sub-grid area |
| Stream order | strord | - | Strahler stream order |
| Elevation | elevtn | m+EGM96 | Hydrologically adjusted outlet pixel elevation where all downstream cells have equal or lower elevation than its upstream neighboring cells, following the algorithm described by Yamazaki et al. (2012) |
| **Meta data** | | | |
| Erroneous flow direction | flwerr | - | Erroneous flow directions (binary), see section 2.4 |
| Upstream area error | upaerr | $\text{km}^2$ | Difference in upstream area error between the upscaled and native resolution river network at the outlet pixel. |
| Outlet pixel coordinates | outlon / outlat | - | Outlet pixel coordinates in EPSG:4326 projection. |

## 2.4 Upscaled flow direction accuracy metrics

We benchmarked the performance of the IHU dataset against DMM and EAM at the global scale, see Appendix A for a
detailed description of these methods. DMM is selected as it is still often applied, for example in the recently published multi-scale routing model (e.g.: Thober et al., 2019), and EAM is selected as it is at the basis of our method. We hypothesize that IHU will show improved results compared to DMM and EAM in areas with many meandering streams and near confluences. IHU is benchmarked based on the following variables:

- *Erroneous flow directions*: The flow direction of a cell is erroneous if the first outlet pixel downstream of the outlet
pixel of that cell is not located in its downstream cell (i.e. where the flow direction points to). Examples of erroneous flow directions are given by the red arrows in Figure 1 panel 2A. This measures the local accuracy of the upscaled flow directions, with less erroneous flow directions indicating a better representation of fine resolution confluences at coarser resolutions.



- *Upstream area error*: The difference in upstream area between the target resolution upstream area at cell $i$ $\widehat{A_i}$ and

the fine-resolution upstream area at the cells' outlet pixel $A_i$. This is an aggregated measure of the accuracy of all upstream flow directions. Based on the upstream area error we define: (2) absolute error $\epsilon$; (3) relative error $\epsilon_{rel}$; and (4) mean relative error $\overline{\epsilon_{rel}}$:

$$\epsilon = \widehat{A_i} - A_i, \tag{2}$$

$$\epsilon_{rel} = \frac{|\widehat{A_i} - A_i|}{A_i}, \tag{3}$$

$$\overline{\epsilon_{rel}} = \frac{1}{N}\sum_{i=0}^{N}\frac{|\widehat{A_i} - A_i|}{A_i}, \tag{4}$$

**2.5 Case study setup**

For a case study of the river Rhine in Europe, we assessed the effect of resolution and upscaling method on simulated river discharge for a synthetic runoff event. For each method we calculated the difference in simulated peak flow magnitude and timing between three upscaling methods at resolution of 30 arcsec, 5 and 15 arcmin and a simulation based on the baseline 3

arcsec resolution. We expect smaller differences for IHU compared to other upscaling methods, especially at river confluences.

The Rhine basin catchment area up to the river outlet near Rotterdam, the Netherlands, has a surface area of approximately 195,000 km², see Figure 3. The basin has many smaller contributing sub-basins including the Meuse basin with its confluence near the river mouth after flowing parallel to the Rhine for many kilometers. Further upstream, the Moselle basin

has many meanders and in the upstream Swiss sub-basins many lakes are present. These features are typically hard to represent at coarser resolutions and therefore allow for a detailed benchmark between upscaling methods. Note that in reality the river flow in the Dutch part of the Rhine is more complicated than can be captured in D8 flow directions with an important bifurcation, splitting the Rhine into the Ijssel and Waal rivers and canals between the Waal and Meuse rivers.

A routing model was setup to simulate channel flow for river cells, here defined as cells with a minimum upstream area of

10 km². Routing was based on a kinematic wave routing model, solved using the Newton-Rhapson scheme as described in Chow et al. (1988) at a fixed timestep of 15 minutes. Runoff of headwater cells and within a river cell is instantly accumulated and fed to the channel at the outlet pixel of that cell. Channel length, slope and width at all resolutions are based on definitions in section 2.2, where for the fine-resolution baseline data every pixel is considered to be an outlet pixel. A default length of 2 km around (1 km up- and downstream of) the outlet pixel was used to calculate the slope. To fill gaps in

the river width observations we fitted a power-law relation between upstream area (A), as a proxy for bank full discharge, and MERIT Hydro river width (w) according to equation 5, where a and b are fitted to be 0.15 and 0.664 respectively, for more details see Appendix B, Figure B1.

$$w = a A^b, \tag{5}$$

In addition, we applied a default manning roughness coefficient of 0.03 and a minimum slope of 0.1 m/km. The default
parameters are selected after a sensitivity analysis of the results to the channel slope length, width and roughness parameters,
see Appendix E, Figure E1-2. While the simulated discharge peak magnitude and timing are sensitive to these parameters in
varying degrees, it does not greatly affect the differences between methods and does not change the conclusions based on it.

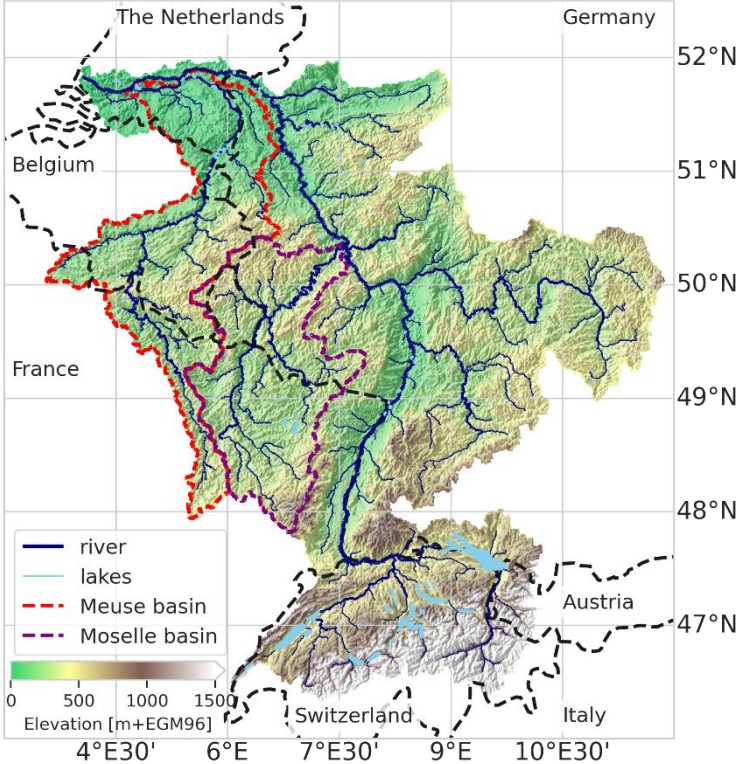

**Figure 3: Study area: Rhine basin with average, elevation, rivers and basin outlines based on MERIT Hydro IHU 30 arcsec**
**dataset (this paper); lakes are derived from the hydro Lakes dataset *(Messager et al., 2016)*.**

## 3 Accuracy of upscaling methods

In this section we benchmark the accuracy of IHU against the DMM and the EAM globally, based on erroneous flow
direction and upstream area errors. Note that the results are presented at different spatial scales, from individual cells (Figure
4) to basins (Table 2) and 1 by 1 degree tiles (Figure 5).

First, we analyze the percentage of native resolution basins that are resolved in the upscaled flow direction maps. A basin is
not resolved when it drains completely into neighboring basin(s) when upscaled and subsequently has no river outlet or pit at
the coarser resolution. At each resolution we analyze the basins with a total area larger than approximately one cell. IHU
resolves more than 96.2% of the basins above the set thresholds compared to 85.7% and 87.6% for DMM and EAM at 30
arcsec resolution while a larger percentage is resolved at courser resolutions, see first row in Table 2. Only 2 basins larger




than 5,000 km² are not resolved at 15 arcmin resolution using IHU. These are an endorheic basin in the south of the Arabian Peninsula (6996 km²) and a small basin in Ontario, Canada (6830 km²), see Figure C1-2. Both are merged with a larger nearby basin as the river mouth or pits runs through the same cell as the outlet or sink and it cannot be not assigned to another neighboring cell. The largest unresolved basins at 5 arcmin has an area of 3521 km² and at 30 arcsec an area of 40 km².

Next, we assess the percentage of cells with erroneous flow directions. This error is at the base of all upscaling errors discussed in this section and thus an important performance metric. The percentage of resolved basins that have less than 5% cells with erroneous flow directions is above 92.2% for IHU at all resolutions analyzed compared to 20.8% for DMM and 43.7% for EAM, see third row in Table 2, indicating that many more confluences are properly resolved at the upscaled resolution. The difference between the methods is smaller for basins with less than 1% cells with erroneous flow directions.

While the second iteration of IHU successfully limits this error compared to DMM and EAM, it cannot be avoided. For cells with parallel fine-resolution flow paths it is impossible to correctly set upscaled flow direction for all cells in the D8 format.

**Table 2: Percentage of resolved basins with area larger than approximately one cell that meet performance criteria based on relative basin area error, relative upstream area error ($\epsilon_{rel}$) larger than 1% and cells with erroneous flow direction. For each criterium, the worst performance across all resolutions per method is highlighted.**

| | 30 arcsec | | | 05 arcmin | | | 15 arcmin | | |
|---|---|---|---|---|---|---|---|---|---|
| | ~1 km² (510637 basins) | | | ~100 km² (27043 basins) | | | ~900 km² (7506 basins) | | |
| | DMM | EAM | IHU | DMM | EAM | IHU | DMM | EAM | IHU |
| 1. Basins resolved (percentage of total basins) | 437821 **(85.7%)** | 447228 **(87.6%)** | 491203 **(96.2%)** | 24060 (89%) | 24693 (91.3%) | 26537 (98.1%) | 6502 (86.6%) | 6758 (90%) | 7336 (97.7%) |
| 2. < 1% cells with flow dir errors | 30.6% | 49.4% | 89.5% | **20.8%** | **42.1%** | **82.2%** | 27.4% | 50.1% | 86.5% |
| 3. < 5% cells with flow dir errors | 30.6% | 50.8% | 95.7% | **20.8%** | **43.7%** | **92.2%** | 27.4% | 51.2% | 92.7% |
| 4. < 1% cells with $\epsilon_{rel}$ > 1% | 27.0% | 45.1% | 85.8% | **16.9%** | **37.4%** | **79.4%** | 23.1% | 45.0% | 84.5% |
| 5. < 5% cells with $\epsilon_{rel}$ > 1% | 69.0% | 79.3% | 95.1% | **59.7%** | **75.4%** | **93.9%** | 69.9% | 83.7% | 96.3% |
| 6. < 1% basin area error | 68.3% | 69.3% | 97.9% | 61.4% | 67.1% | 96.6% | **61.0%** | **66.0%** | **96.3%** |
| 7. < 5% basin area error | 75.0% | 77.1% | 98.3% | 70.5% | 76.2% | 97.2% | **68.1%** | **74.9%** | **96.8%** |


The relative upstream area error ($\epsilon_{rel}$) considers the cumulative upstream error of erroneous flow directions. We find that more than 93.9% of the resolved basins have less than 5% cells exceeding the 1% upstream area error threshold for IHU compared to 59.7% for DMM and 75.4% for EAM, see fifth row in Table 2. The difference between the methods is larger for basins which have less than 5% cells exceeding the 1% upstream area error threshold. Figure 5 shows the percentage of

cells within a 1 by 1 degree tile with a relative upstream area error larger than 1% for 5 arcmin resolution output maps. The differences between methods are consistent across the resolutions analyzed, see Figure D1-2. For IHU, most tiles have less than 1% cells with larger than 1% relative upstream area error. Exceptions are found in mountainous, glacierized and dry-land regions, see green areas in Figure 5. For example, in dry-land areas such as the south part of Arabian Peninsula, North part of Lake Eyre in Australia and some parts in the Sahara, where large rivers are absent, existing streams are ephemeral

and flow directions extremely parallel, see for example Figure C1 and C4. In regions covered by ice sheets such as Greenland, streams are not well depicted by the terrain elevation based on which flow direction are estimated to be extremely parallel. In such areas, even at high resolutions, flow directions are highly uncertain.





The basin area error is analyzed based on the relative upstream area error at the basin outlet cell, as shown with dots in Figure 5 for the 500 largest basins globally. For IHU more than 96.8% of the resolved basins have a basin area error relative

to original basin area of less than 5% compared to 68.1% for DMM and 74.9% for EAM, see seventh row in Table 2. The difference between the methods is larger for the basins with a total area of less than 1%. While the 4$^{th}$ iteration of IHU, see section 2.1, successfully limits this error in comparison to DMM and EAM it cannot completely be avoided. Large basin area errors of more than 100 km$^2$ for basins larger than 1000 km$^2$ (10%) occur at 360 basins at 15 arcmin, 91 at 5 arcmin and zero at 30 arcsec resolution. The largest basins with a 10% relative basin area error at each resolution are shown in Figure C3-5.

Sections of basins can still be merged with neighboring basins when a cell gets isolated between cells from another basin. This occurs when none of the neighboring cells share a downstream outlet pixel.

The absolute upstream area error for all cells shows an improvement in performance for IHU compared to DMM and EAM, see Figure 4. While the error increases slightly with lower resolutions it is consistently lower at all resolutions for IHU. At the 5 arcmin resolution 2.5% of the cells have a positive and 0.7% a negative upstream area error compared to 9.9% positive

and 30.2% negative for DMM and 15.3% positive and 5.8% negative for EAM. In general, DMM shows a large percentage of negative upstream area errors while EAM and IHU tend to be skewed towards positive errors. Negative errors typically result from upscaled flow directions that cause a shortcut in a meandering section of a stream. The cells between the start and end of the shortcut then become a new branch in the upscaled flow direction map with smaller upstream area. Both positive and negative errors occur when a confluence in the upscaled flow directions is erroneously located upstream from the actual

confluence, thereby increasing the upstream area in one branch while decreasing it in the other branch where the number of cells with a positive or negative error depends on the length and the number of outlet pixels on each branch, see example in Figure C6.

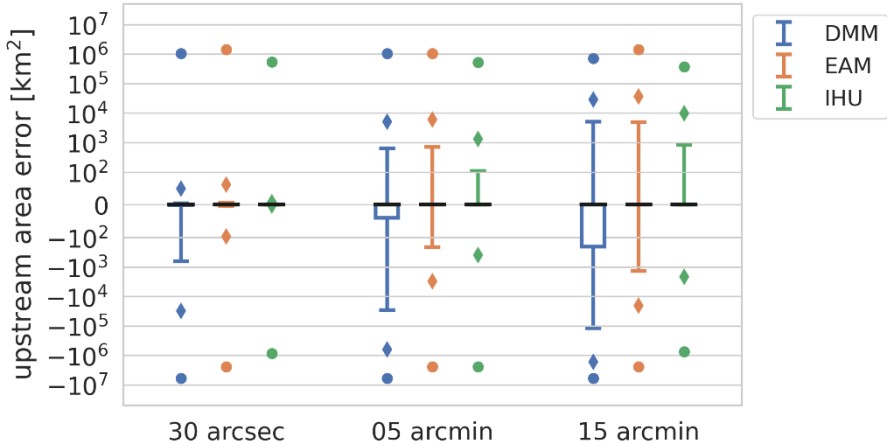

**Figure 4: Absolute upstream area for DMM ( blue), EAM (orange) and IHU (green) at three different resolution from 30 arcsec**
**(~1km; left column) to 15 arcmin (~30 km; right column), where the black lines indicate the median error, the box the 25 -75 percentiles, the whiskers the 1-99 percentiles, the diamonds the 0.1-99.9 percentiles and the dots the min and max errors.**





**Figure 5: Relative upstream area error ($\epsilon_{rel}$) at a 5 arcmin resolution (~10km) for DMM (upper), EAM (middle) and IHU (bottom). The background colors show the percentage of cells per 1x1 degree tile with a relative upstream area error of more than 1%, while the markers show the relative upstream area error at the outlets or sinks of the 500 largest basins globally (black lines).**





# 4 Effect of upscaling method on simulated discharge

In this section we assess the effect of the flow direction upscaling method on simulated discharge for a case study of the river Rhine basin. First, we discuss the upscaled flow direction maps with resulting upstream area error as shown in Figure 6.

Compared to DMM and EAM, the upstream area error for the Rhine basin based on IHU is smaller (negligible at 30 arcsec) and more localized. A clear error that occurs at 5 and 15 arcmin resolutions with DMM and EAM is the erroneous confluence of the Meuse river which is merged in the main stem upstream from the actual confluence, see Figure 6. Furthermore, at 30 arcsec and 5 arcmin resolution many meanders in the Moselle basin are not correctly resolved with DMM and EAM. For IHU at 15 arcmin a small error in the total basin area is made as a small stream near the outlet is erroneously

merged into the Rhine basin yielding a small error of 55 km$^2$ (0.02%) at the outlet, see Figure 6I.

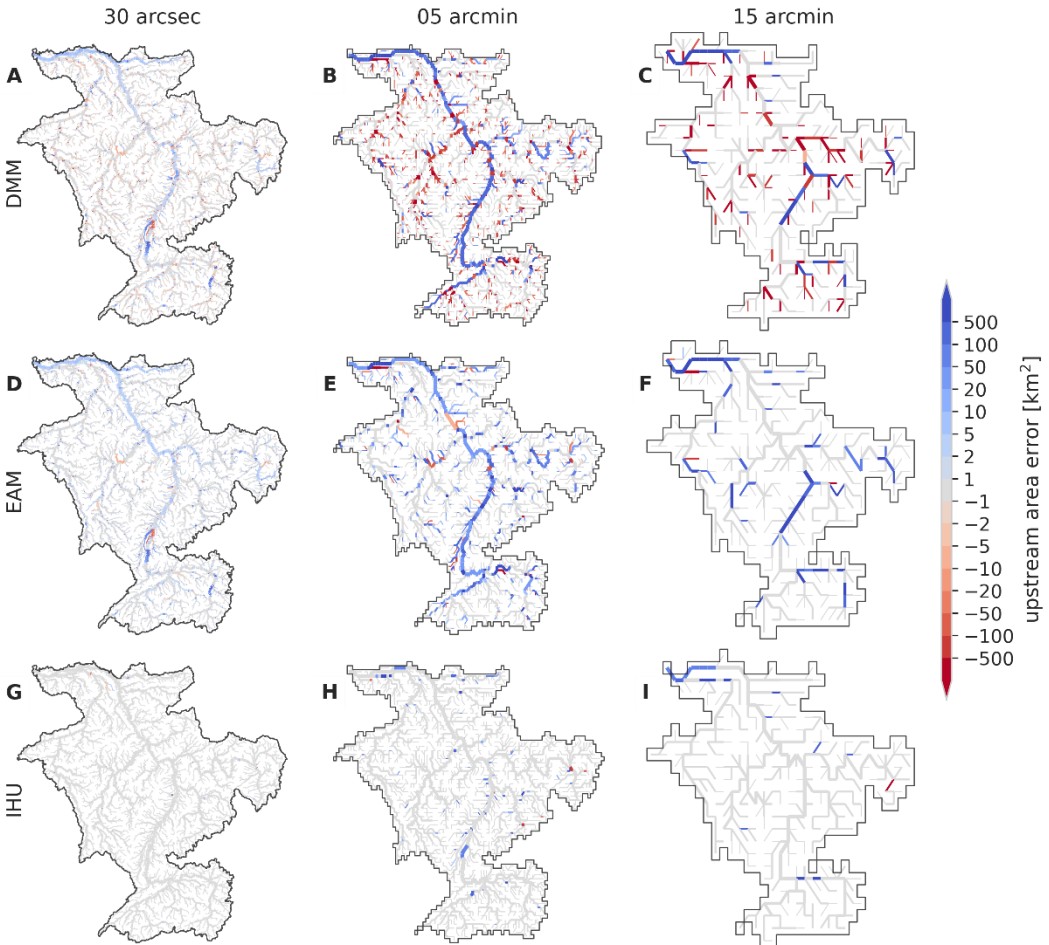

**Figure 6: Upscaled MERIT Hydro flow direction network for the Rhine river at resolutions of 30 arcsec (left column), 5 arcmin (center column) and 15 arcmin (right column) as derived with DMM (first row), EAM (second row) and IHU (third row), where red colors indicate a negative and blue colors a positive upstream error. The line thickness is scaled with the upstream area.**





These flow direction maps together with sub-grid drainage area and river map are used to setup a distributed routing model

to simulate discharge as the result of a synthetic runoff event that is uniformly distributed throughout the catchment. We

analyze the difference between simulated discharge in the upscaled model compared to the original 3 arcsec model. Figure 7

shows the runoff event (left y-axis) and resulting flood peak wave at the river outlet (right y-axis) for all methods and

resolutions. It is directly clear that models based on EAM (orange) and IHU (green) perform much better, i.e. show better

similarity to the model at the original 3 arcsec resolution, than the models based on DMM (blue). The largest error in flood

magnitude (+468 $m^3s^{-1}$) and largest error in flood peak timing (-54 hours) are found for DMM at 5 and 15 arcmin resolution.

These errors are likely due to a positive total area error in combination many shortcuts in the upscaled river network. The

largest errors in flood peak magnitude for EAM (-299 $m^3s^{-1}$) and IHU (-268 $m^3s^{-1}$) are found at 15 arcmin resolution and the

largest error in flood peak timing for EAM (+9 hours) and IHU (+7 hours) are found at 15 and 5 arcmin resolution. Both

errors for both models have opposite sign compared to models based on DMM. These errors are likely due to the smoothing

effect of the longer river channels at coarser resolutions. While there are clear differences in the upstream area error between

EAM and IHU, see Figure 6, the differences in simulated flood peak at the river outlet between EAM and IHU are small.

This is likely because the effect of upstream area errors on simulated discharge cancel out at the river outlet resulting in a net

similar effect on the simulated discharge.

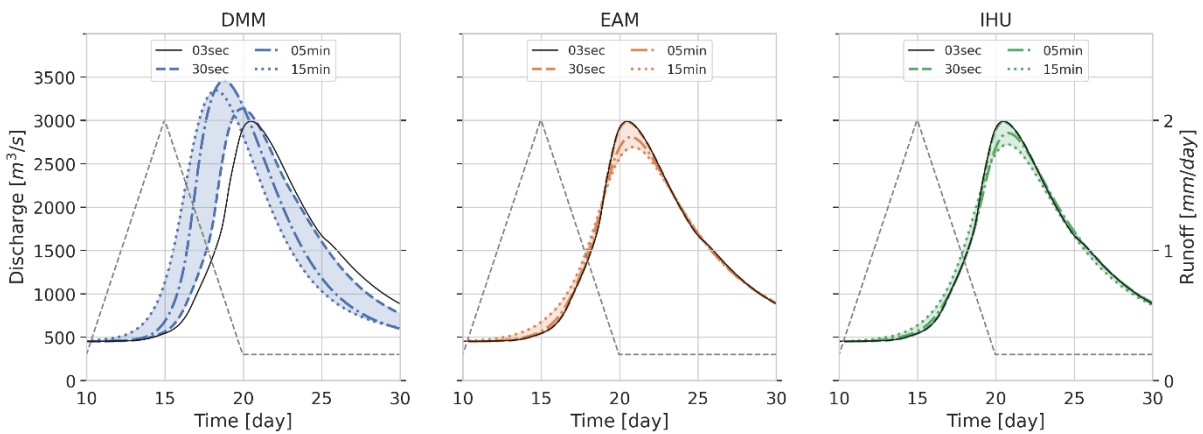


**Figure 7: Simulated discharge at the river mouth of the Rhine river near Rotterdam for a synthetic runoff event (grey line) and based on models with native 3 arcsec resolution flow directions (black line) in comparison to upscaled flow directions based on DMM (blue; left), EAM (orange; center) and IHU (green; right) at 30 arcsec (dashed line), 5 arcmin (dash-dotted line) and 15 arcmin (dotted line) resolution. Note that the simulated discharge for the 30 arcsec EAM and IHU runs largely overlap with the**
**native 3 arcsec run.**

To better understand the effect of flow direction upscaling on river routing we therefore extend this numerical experiment to

many locations across the Rhine basin, see Figure 8. The locations are selected based on the outlet pixels at 15 arcmin

resolution that are at the same location or near outlet pixels a higher resolution. We use a maximum relative upstream area

error of 1% to select nearby outlet pixels at higher resolutions on the same stream. For EAM these are far less than for DMM

and IHU as the outlet pixel locations are selected within the effective area rather than at the edge of a target cell and

therefore less likely to be the same between resolutions. Generally, simulated discharge based on the IHU upscaled river





network yields smaller flood peak magnitude and timing differences compared to DMM and EAM. More locations with a flood peak magnitude smaller than 2.5% are found for IHU (41.0%) compared to EAM (26.3%) and DMM (17.7%) across all resolutions, see white dots in Figure 8. Similarly, more locations with a difference in flood peak timing smaller than 2 hours are found for IHU (21.2%), compared to EAM (12.5%) and DMM (5.9%), see white dots in Figure 9. In general, the IHU models show better similarity to the native resolution model across the full distribution of locations for both flood peak magnitude and timing, see thick black lines in Figures E1-2. The differences in simulated discharge are caused by (local) upscaling errors in the river network. Both in the DMM and EAM low resolution river networks the Meuse River (most downstream and largest tributary with a size of about 16% of the total basin) is merged into the main stem upstream from the actual confluence, causing large differences in both flood peak timing and magnitude in the Meuse section downstream of the erroneous confluence. Differences in flood peak simulation between upscaled and native resolution models are not only due to upscaling errors, also the upscaling of river width and slope are crucial for scale invariant discharge simulations. For instance, differences in river width between resolutions in the upstream part of the Rhine Basin cause a double peak at high resolutions ($\leq$ 30 arcsec) to smoothen out into a single peak at lower resolutions, yielding a large difference in flood peak timing, see Figure 9F and I. In the model, lakes and reservoirs are implicitly represented by larger channel widths from the MERIT Hydro dataset, introducing a buffering capacity in the river system. As the river width at lower resolutions is represented by the width at the outlet pixel of each cell the buffering capacity is different. Averaging the river width would change the buffering capacity as this results in a smoothened river width. The sensitivity of the simulated flood peak timing to river width is shown in the first row of Figure D2, where simulated river width, see Appendix B, is used for gaps in the river width data only (default); for lakes and data gaps or for all cells. Using simulated river widths at lakes and reservoirs, minimized the timing error in the (upstream) Rhine. Besides width, the error in flood peak timing is also sensitive to the length along which the river slope is calculated, see Figure D2. If this length is varied with the model resolution, large errors in flood peak timing are introduced. However, the flood peak timing error is less sensitive to the precise length within a range of 1 to 5 km.




**Figure 8: Difference in simulated flood peak magnitude (ΔQ$_{max}$) in the Rhine river basin between resolutions of 30 arcsec (left column), 5 arcmin (center column) and 15 arcmin (right column) and the baseline 3 arcsec resolution as upscaled with DMM (first row), EAM (second row) and IHU (third row). N denotes the number of observation grid points with comparable outlet pixels locations across all dimensions which are selected based on a maximum 1% upstream area difference. The marker size is scaled with upstream area and markers are plotted in order with increasing upstream area.**






**Figure 9: Difference in simulated flood peak timing ($\Delta T_{Q_{max}}$) in the Rhine river basin between resolutions of 30 arcsec (left column), 5 arcmin (center column) and 15 arcmin (right column) and the baseline 3 arcsec resolution as upscaled with DMM (first row), EAM (second row) and IHU (third row). N denotes the number of observation grid points with comparable outlet pixels locations across all dimensions which are selected based on a maximum 1% upstream area difference. The marker size is scaled with upstream area and markers are plotted in order with increasing upstream area.**

## 5 Discussion

The proposed IHU upscaling method was shown to successfully upscale flow direction data from 3 arcsec data to resolutions up to 15 arcmin. IHU balances between traditional methods such as DMM and EAM which only use local information and DTR which uses global information about the hierarchy of streams to determine which sub-grid stream to represent in each cell. IHU makes a first estimate of the representative sub-grid stream, but updates this for cells with erroneous flow





directions based on contextual information. This makes IHU better suitable to be applied to high resolution hydrography data.

Compared to EAM and DMM, flow directions are better resolved, specifically near confluences. This is important for
correctly modelling flood peak propagation downstream of confluences, especially when flood peaks in nearby (sub-)basins are correlated (Berghuijs et al., 2019). Erroneous IHU upscaled flow directions are still found in dry-land and ice-covered areas where the actual flow directions are also highly uncertain. These upscaling errors are mainly caused by many parallel flow paths in the fine resolution hydrography data. This is partly a limitation inherent to the D8 format, which cannot represent multiple rivers that run parallel through a cell, making it impossible to upscale flow direction data without any
errors. While a large majority of the basins has very small total area errors up to 15 arcmin resolution if upscaled with IHU, the small number of large basins with more than 10% area error does increase with decreasing resolution. The exact upper limit of tolerable upscaling errors and thus upscaling resolution, depends on its application and region of interest. We believe that the 15 arcmin maps are suitable for many global scale applications but results in reported areas with lower accuracy should carefully be interpreted. To guide the user on the quality of the upscaled MERIT hydro IHU dataset we therefore
provide qualitative metadata of erroneous flow direction and upstream area error. To overcome flow direction errors at coarse resolutions, the Flexible Location of Waterways (FLOW) method by Yamazaki et al. (2009) uses a format that allows a downstream cell to be located outside the eight direct neighbors to circumvent this problem. While proven to be effective, most distributed hydrological models continue using the D8 format. The DTR method by Wu et al. (2011) tries to solve the parallel flow path issue by allowing for rivers to be diverted through adjacent cells in favor of smaller rivers within that cell.
Potentially, a stepwise upscaling procedure would even better preserve the largest basins and could be an interesting avenue for further research.

Besides flow directions, IHU derives additional layers of sub-grid drainage area, river length, width and slope data and hydrologically adjusted elevation, which are required for most distributed routing models. Several studies have shown that it is very hard to calculate reliable riverbed slopes from global DEMs (LeFavour and Alsdorf, 2005), while river routing based
on a kinematic wave solution, as commonly used in large scale hydrological models, is very sensitive to the river slope (Thober et al., 2019; Yamazaki et al., 2011). We found that a constant length across all resolutions of at least 1 km (500 m up- and downstream of the outlet pixels) is required to provide a relatively scale invariant estimate of river slope as applied in the case study, see Figure E1-2. To achieve complete coverage of river width, the sub-grid river width data requires to be interpolated for data gaps and lake and reservoir areas if these were to be modelled explicitly in the routing model. Here, we
used a simple power-law relationship between width and upstream area that works well for smaller basins in temperate climate zones. For larger basins in other climate zones, this estimate should be improved using the well-established geomorphic relationships between bank-full discharge and river depth as proposed in the downstream hydraulic geometry framework (Leopold and Maddock, 1953), for instance by the clustering approach proposed by Andreadis et al. (2013) or additional river width data from e.g. Allen and Pavelsky (2018).





In this study we benchmarked several upscaling methods based on the same baseline hydrography data and based on a synthetic runoff event. These choices were made to control the experiment in order to focus on differences in upscaling methods. For future studies it would be interesting to also compare the accompanying dataset with often used hydrography datasets at 30 arcsec resolution such as hydroSHEDS (Lehner et al., 2008) and hydro1k (U.S. Geological Survey, 2000), both in terms of accuracy of the river network and effect on simulated discharge for actual events.

Most multi-resolution routing models use data at pre-processed resolutions (Li et al., 2013; Wu et al., 2014; Yamazaki et al., 2011). However, Thober et al (2019) recently presented a multi-scale routing model that includes the upscaling of flow direction data based on DMM. Using a multi-resolution modelling approach based on IHU could potentially allow for models calibrated on coarser scales than the actual model application to reduce computational expense, especially when model parameters are based on global transfer functions (Imhoff et al., 2020; Samaniego et al., 2017). Furthermore, the

model resolution could be varied within the model domain to add resolution where required.

## 6 Conclusions

To describe flow directions and sub-grid river parameters in distributed hydrological models of different resolutions based on increasingly higher resolution hydrography datasets, automatic upscaling methods are required. The Iterative Hydrography Upscaling (IHU) method takes high resolution flow direction data and upstream area data as input to derive

flow directions at a coarser resolution while preserving the river structure. IHU was successfully applied to the 3 arcsec MERIT Hydro dataset to derive the MERIT hydro IHU multi-resolution hydrography dataset at resolutions of 30 arcsec (~1km), 5 arcmin (~10 km) and 15 arcmin (~30 km). Additional layers of sub-grid drainage area, river length, slope and width parameters are derived based on fine-resolution elevation and river width data.

Compared to other often used upscaling methods, upscaled flow direction maps with IHU show improved accuracy on all

metrics presented globally. For a case study of the Rhine basin we show that using IHU based hydrography maps allows to use lower resolution routing models with similar result for the entire the basin. Besides the upscaled flow direction data, the model similarity is also sensitive to how sub-grid river slope and width variables are derived. Because IHU is completely automated and yet accurate, it allows for a rapid uptake of new high-resolution flow direction data in distributed hydrological models at different resolutions.






**Appendix A: Upscaling methods used for benchmarking**

The EAM, DMM and IHU have three steps in common. First, a pixel is selected for every cell that determines the representative river within the cell. Second, the pixel is traced downstream until a certain criterium in a neighboring cell is met. Third, the upscaled flow direction is set towards that neighboring cell. The differences between the methods are based

on how the first two steps are implemented. DMM and EAM are illustrated in Figure A1 and briefly describe below, more detailed descriptions can be found in the papers referenced.

**Double maximum method:**

- Step 1A: The outlet pixel is defined as the pixel with the largest upstream area that is either a basin outlet pixel within the cell or located at the edge of the cell (grey squares).

- Step 1B: The cell is offset half the cell size in the direction of the cell quadrant in which the outlet pixel is found (dashed grid lines). The outlet pixel is then traced downstream until it leaves the offset cell (black lines).

- Step 1C: The upscaled flow direction is set to the cell where the trace in step 1B ends (orange arrows).

For a detailed description of the method we refer to Olivera *et al* (2002).

**Effective Area Method:**

- Step 2A: The representative pixel (dark red squares), is defined as the pixel with the largest upstream area which is located within the effective area (shaded area) defined by equation 1, see section 2.1.

- Step 2B: The representative pixel is then traced until the first downstream effective area is reached, which by definition is in a neighboring cell (black lines).

- Step 2C: The upscaled flow direction is set to the cell where the trace in step 2B ends (orange arrows).

For a detailed description of the method we refer to Yamazaki *et al* (2008).

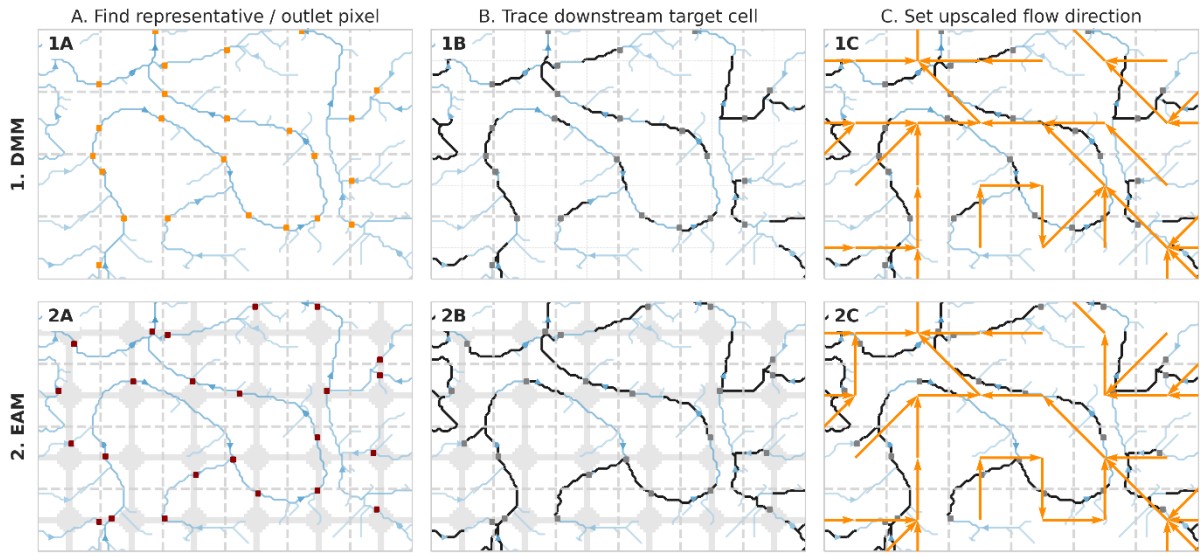





**Figure A1: Visual explanation of the Double Maximum Method (DMM; first row) and Effective Area Method (EAM; second row). The target resolution grid (grey lines) and fine-resolution upstream area map (blue colors) are shown in all plots. The representative (EAM) or outlet (DMM) pixels (squares) are traced downstream until a criterium in a neighboring cell is met (black lines) and the upscaled flow direction are set (orange arrows).**

**Appendix B: River width interpolation**

To fill gaps in the river width observations we fitted a power-law relation between upstream area (A), as a proxy for bank full discharge, and MERIT Hydro river width (w), see equation 4.The MERIT Hydro river width was taken for all river cells within the Rhine catchment excluding cells within lakes and reservoirs based footprints from the hydro lakes (Messager et al., 2016) and GRAND databases (Lehner et al., 2011). We used a least squared error fitting algorithm from the python scipy.optimize package (Virtanen et al., 2020) which was iteratively fitted to the sample after removing outliers based on the best fit. Outliers are defined based on the difference between observed and simulated width of at least 200 m (for small widths) and the simulated width (for widths larger than 100 m).

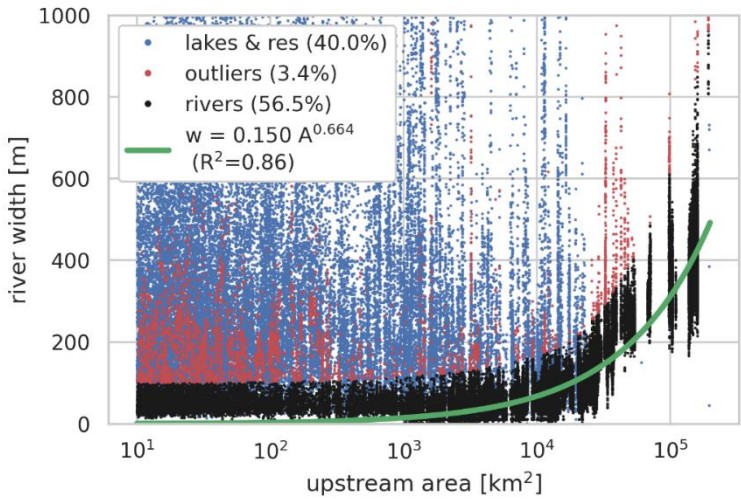

**Figure B1: Fitted relationship between river width and upstream area for the river Rhine basin**





# Appendix C: Examples of upscaling errors

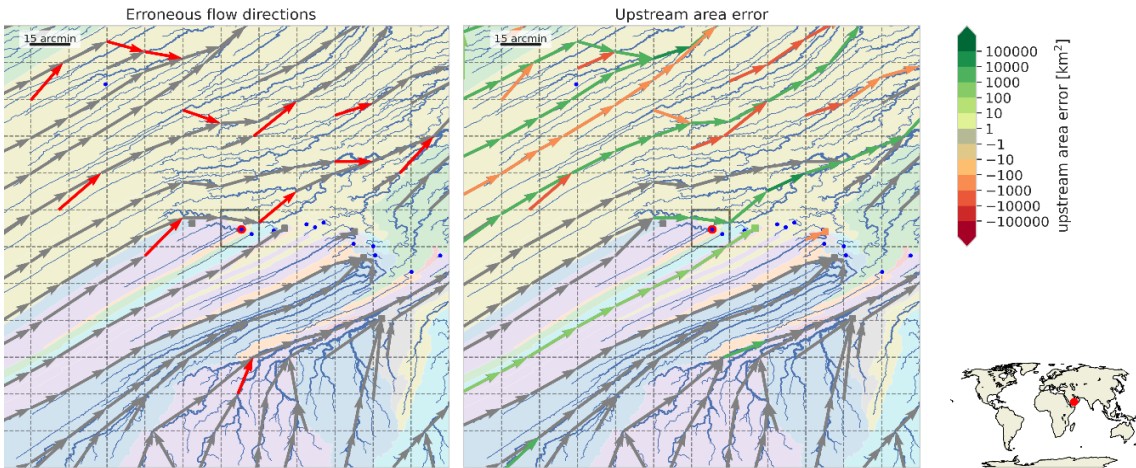

**Figure C1: Largest endorheic basin (6996 km²) which is not resolved at 15 arcmin resolution indicated with highlighted basin pit.**
**The blue lines show the fine resolution river, the background colors show basin boundaries, the dash lines the coarse resolution grid and the arrows the upscaled flow directions pointing from the outlet pixel or the original cell to outlet pixel of the destination cell. Flow directions are red if erroneous (left) and green for a positive- and red for a negative upstream area error (right).**

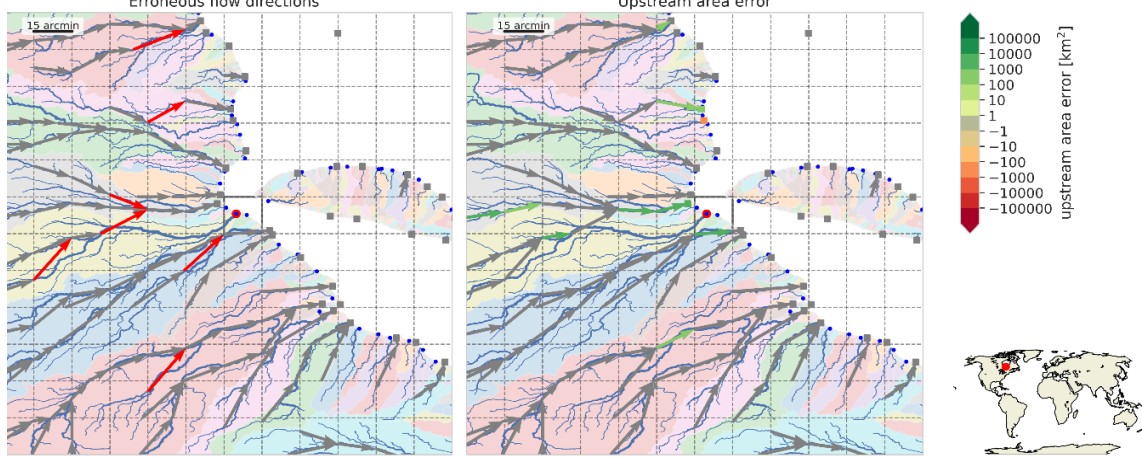

**Figure C2: Largest exorheic basin (6830 km²) which is not resolved at 15 arcmin resolution indicated with highlighted basin outlet.**
**See caption of Figure C1 for a full description.**



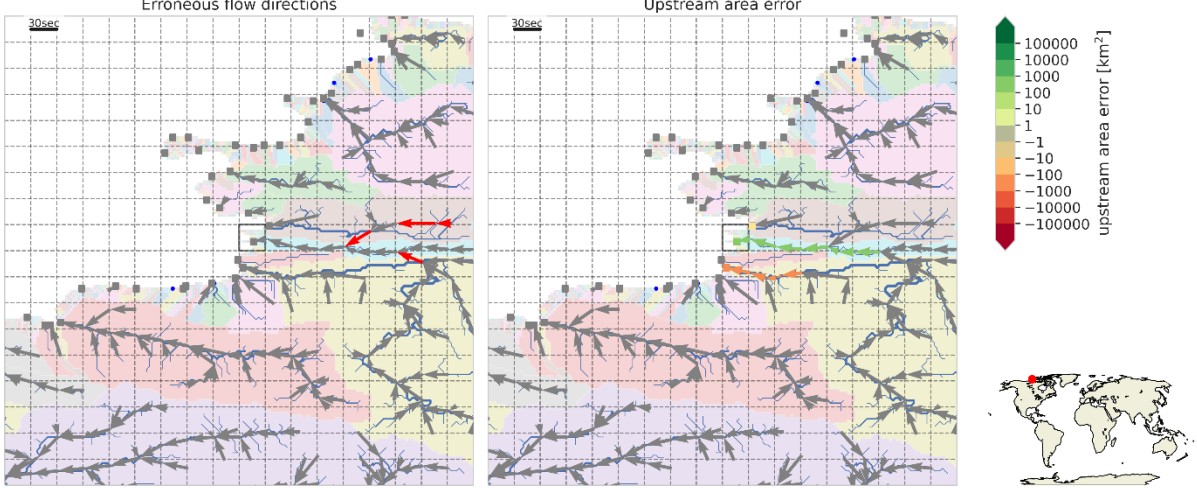

**Figure C3: Largest basin (914 km²) with a relative upstream area error of more than 10% at 30 arcsec resolution indicated with highlighted cell. See caption of Figure C1 for a full description.**

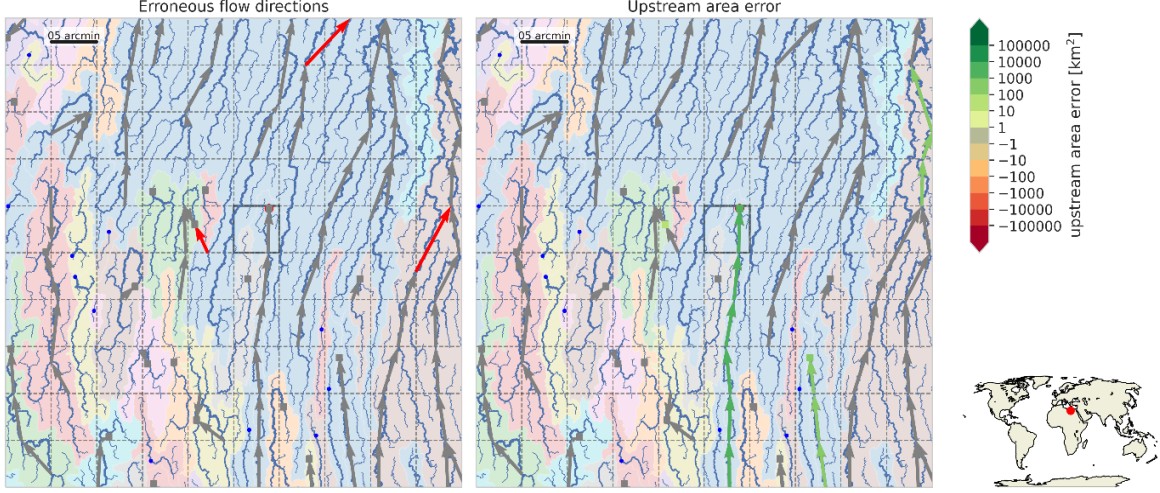

**Figure C4: Largest basin (16717 km²) with a relative upstream area error of more than 10% at 5 arcmin resolution indicated with highlighted cell. See caption of Figure C1 for a full description.**





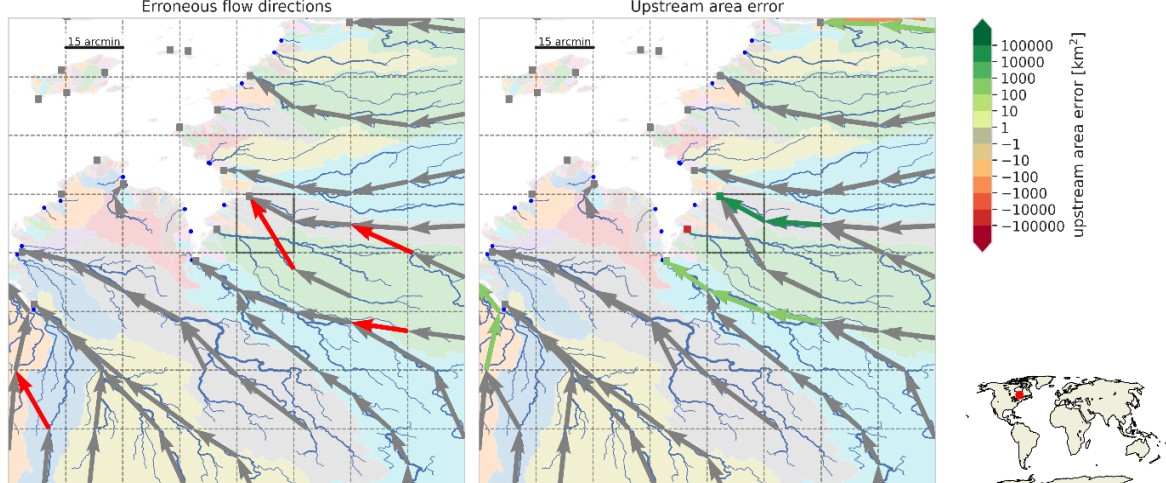

Figure C5: Largest basin (42017 km$^2$) with a relative upstream area error of more than 10% at 15 arcmin resolution indicated with highlighted cell. See caption of Figure C1 for a full description.

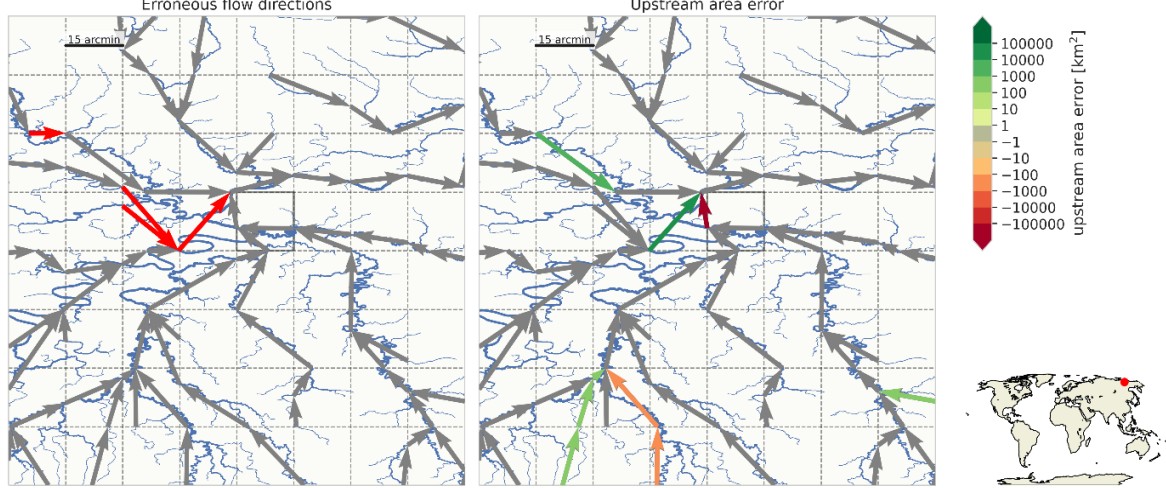


Figure C6: Second largest local upstream area error (-220876 km$^2$) at 15 arcmin resolution indicated with highlighted cell. See caption of Figure C1 for a full description

## Appendix D: Accuracy benchmark of upscaling methods

This section shows maps of the relative upstream area error at resolutions of 30 arcsec and 15 arcmin in addition to the map

at 5 arcmin as presented in section 3.



**Figure D1: Percentage of cells at a 30 arcsec (~1km) resolution per 1x1 degree tile with an absolute relative upstream area error of more than 1%, while the markers show the upstream area error at basin outlet and the black lines the outlines of the 200 largest basins globally.**




**Figure D2: Percentage of cells at a 15 arcmin (~30km) resolution per 1x1 degree tile with an absolute relative upstream area error of more than 1%, while the markers show the upstream area error at basin outlet and the black lines the outlines of the 200 largest basins globally.**





## Appendix E: Sensitivity analysis

The sensitivity of the model similarity between upscaled and baseline resolution models to three key model variables is presented in this section. The similarity is expressed as difference in flood peak timing and magnitude between the upscaled and native resolution model.

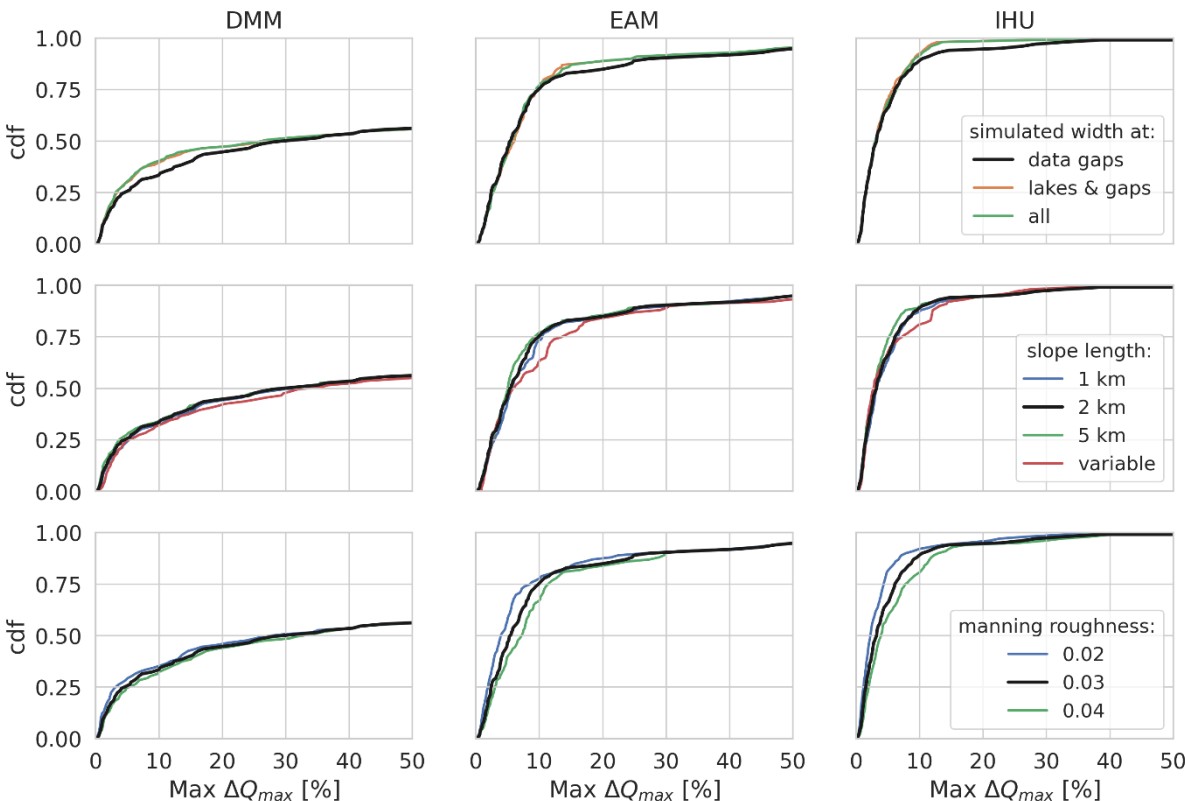

**Figure E1: Sensitivity analysis of relative difference in simulated peak magnitude for a case study in the Rhine basin to three**
**parameters (rows) and for three upscaling methods (columns). Each plot shows the CDF of all output location on the y-axis and the relative difference in simulated peak magnitude on the x-axis. The first row shows the sensitivity to the average MERIT Hydro vs power-law based channel width estimates; the second row shows the sensitivity to the minimum channel length over which the channel slope is estimated, and the third row shows the sensitivity to the manning roughness coefficient. The black line is the default case in all plots.**





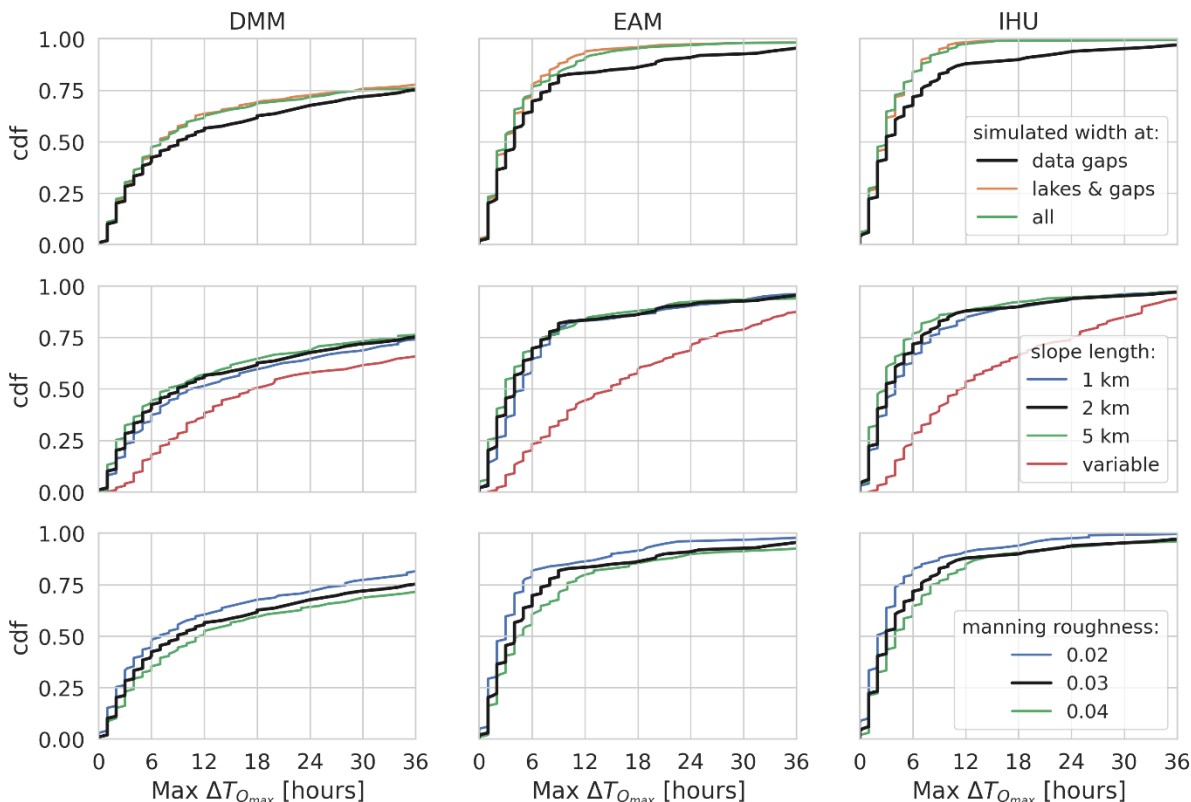

**Figure E2:**
**Sensitivity analysis of relative difference in simulated peak timing for a case study in the Rhine basin to three parameters (rows) and for three upscaling methods (columns). Each plot shows the CDF of all output location on the y-axis and the relative difference in simulated peak magnitude on the x-axis. The first row shows the sensitivity to the average MERIT Hydro vs power-law based channel width estimates; the second row shows the sensitivity to the minimum channel length over which the channel slope is estimated, and the third row shows the sensitivity to the manning roughness coefficient. The black line is the default case in all plots.**



### Author contribution

The IHU algorithm was developed implemented in pyflwdir by D.E. in close collaboration with W.V. and D.Y. The experiment was designed by D.E., P.W., H.W. and A.W and executed by D.E. All authors contributed to the manuscript.

### Competing interests

The authors declare that they have no conflict of interest.



**Code and data availability**

All upscaling algorithms used in this study are implemented in the open-source python *pyflwdir v0.4.4* package (https://pypi.org/project/pyflwdir/0.4.4/)

The multi-resolution *MERIT hydro IHU* dataset is available for download from zenodo.org (https://doi.org/10.5281/zenodo.4138776), under CC-BY NC or ODbL v1.0 license.

**Acknowledgments**

The research leading to these results received funding from the Dutch Research Council (NWO) in the form of a VIDI grant (grant no. 016.161.324), the IMPREX project, as funded by the European Commission under the Horizon 2020 framework

Program (Grant 641811) and by the Dutch Ministry of Infrastructure and the Environment. The contribution by D.Y. is funded through "Cross-ministerial Strategic Innovation Promotion Program" of Japan.

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
