# Peer review of "A hydrography upscaling method for scale invariant parametrization of distributed hydrological models"

_Hydrology and Earth System Sciences, 2020_

## Referee Comment (RC1) · Anonymous Referee #1 · 22 Dec 2020

**Referee comment**

**A hydrography upscaling method for scale invariant parametrization of distributed hydrological models**
Dirk Eilander, Willem van Verseveld, Dai Yamazaki, Albrecht Weerts, Hessel C. Winsemius, and Philip J. Ward

The authors describe a method to uspscale high-resolution data, based on a back-tracking approach also used by former work of Yamazaki et al. 2009 or Wu et al. 2011). The authors show methods to assess the quality of different methods.

The paper represents significant progress as it is a further development of the concept of Yamazaki et al:

- The authors can show that there approach yields better results in comparison to other methods
- It is open source
- The authors show that their method can be applied in hydrological models towards free scalable models
- It can be used as the common D8 network

The presentation quality and scientific quality is good. A few points could be discussed in a different way.

Main point of criticism is that it does not include a link to the work done in the ISI-MIP (https://www.isimip.org/) project. In this project, quite a number of hydrological models use a defined set of input data as a global 30 arcmin setting. The network used here is the DDM30 (Doell and Lehner 2002). It is questionable, if this database is the best choice (see Zhao et al. 2017 https://doi.org/10.1088/1748-9326/aa7250), but it is used as the defined river network. The paper can run without a direct comparison to DDM or a comparison to 30 arcmin, but the value of this paper (and the numbers of citations) can be improved, if it is compared against:

a.) 30 arcmin (maybe instead of 15 arcmin, which is rarely used in hydrological models)
b.) DDM (maybe instead DMM, as the DMM is not so often used (cited 25 times and DDM cited 147 times)

Using a power function of upstream area for river width is a weak point here. This does not work for a global dataset and not even for the River Rhine with high runoff in the mountains and low runoff in the lowlands. This approach is not state of the art. The paper says it will provide a parametrization for distr. hydrological models. The approach for providing river width is not appropriate. For sure it would be fine to have the full package incl. river width and Manning's roughness. I think it is still a fine paper, if you exclude river width (as you exclude Manning's anyway)

**In detail comments:**

60: As I said, I am not a fan of the DDM30, but it is THE reference river network in a global hydrological intercomparison project.

66: Wu et al. 2011,2012 and Yamazaki (2009) already give out length, Yamazaki already give out slope or elevation at the outlet point.

85 .. often defined by … Isn't it a requirement to be a multiple of the finer grid?

115: the equation needs $|x-x_0|^{0.5}$ instead of brackets

118: if no output pixel is found… where does this happen in fig 1

126: Maybe a description like in chess B2 instead h would be easier

154: You have several thresholds in your method e.g. sqrt(R), min upstream_area = 0.25, length of cell = 0.25. Did you test this setting, did you do a sensitive analysis of these values. Where are they from?
Maybe for the Rhine it would be good to show some variation of these thresholds

189: Nice solution of these "orphan" problem – upstream cells that have no direct parents

204-206: This paper does not show a valid way to derive river width for all cells (and it is stated well, you need the parameter for all cells). Filling up with a power function of upstream area will not work. Maybe using some regression/machine learning technique like in Barbarossa et al. 2018 (https://www.nature.com/articles/sdata201852) will help. Maybe dismiss 204-206 and 2F (and write a second paper on width and Manning's). For the routing example, your assumption of width and Manning's is ok

205: Which outlet pixel in @F does not have a river width in fig 2F. Using river width interpolation with a power function and upstream area is a really weak assumption.

222: As before: it is a dataset, but an incomplete dataset for kinematic routing missing an adequate solution for river width and channel roughness. I am not asking for these 2 additional parameters, but mentioning that these 2 are missing for a complete dataset (and maybe river depth, too).

229: Also mentioned before: DMM is rather special, a comparison to DDM30 would be better

260: Not so clear, why the minimum upstream area is chosen to be 10 km2. Is this done only for the original 3arcsec, or also for the 30 arcsec version? A 1km2 threshold would be more reasonable? Why having a network to 30 arcsec and then aggregating again to 10km2? Adding a reason for 10km2 would be ok.

265-273: For the synthetic runoff event, a simple assumption is ok, like equation 5 and roughness=0.03. What about the river depth to get the perimeter you need for the kinematic routing?

273: later in357 you describe the synthetic runoff event, It would be better to describe here in more detail how your synthetic event looks like. From your sentence in line 356 I assume: Uniform

runoff for the whole Rhine of around 0.2 mm per day (0.002 per 15 min) to reach 500 m/s at run outlet and then increasing to 0.014 mm per 15 min to reach peak of 3000 m/s?

350: A table for the Rhine as the table 2 would be good. Maybe even later and including the synthetic runoff results.

371: grey line as cumulated runoff?

376-404: This part is interesting, but would benefit if it concludes in a method which can be used to compare different methods in numbers e.g. creating a table with flood peak magnitude (btw a flood at around 3000 m3/s is not a flood in the upper Rhine) and timing like the numbers in line 383 and line 385.
But due to the different N, the numbers are not really comparable. How about selecting only locations which all methods have in common, describing a method to find these locations and then it is possible to set up a table with peak magnitude smaller or bigger than 2%, 5%, 10% and flood peak timing different by percent of runoff peak time to routing peak time

384: maybe not an absolute hour, but a percentage of the difference between running time between runoff peak (gray line) and reference time (3 arcsec model). Because it does not matter so much if the delay in a big basin is 2 hours, but it matters if it is in a small basin.

442  As before: river width I would delete from this list as you cannot derive it in a proper way with IHU.

448-444: To my understanding, there is no well-established geomorphic relationship between discharge and river depth nor river width. It is mostly regression between discharge, upstream area, etc. like in Pistocci 2006 or others. A way to improve this is machine learning and feeding it with climatic, geomorphic data like Barbarossa et al. 2018 or using advanced technics of remote sensing e.g., Allen and Pavelsky 2018 or a combination of both. Your approach for having a full dataset for benchmarking is ok, but maybe dismiss the part for river width.

465: The Merit DEM (Yamazaki et al. 2017) is a very good DEM. But mostly due to anthropogenic overprinting, rivers do not always follow the lowest elevation. They are redirected or even on a higher level than the surrounding landscape (there are many examples of this in the Netherlands). One way to improve the DEM would be to burn in the existing river network. Not part of your work here, but maybe something to think of in the discussion.

---

## Referee Comment (RC2) · Anonymous Referee #2 · 16 Mar 2021

I am sorry I took so long for reviewing this paper. This is not due to the topic the paper deals with but the way the paper is organised. I do not want to appear definitive, however, if the topic, of adapting the representation of a river network at certain resolution to be the more effective as possible, is of geographical interest, I am still not convinced it is also of hydrological importance. The Authors should work a little more to make the paper more appealing, instead they buried the good ideas under a lot of technicalities which are, certainly, important but secondary to the reasons why they do it. Finally some light came in with Figure 7 and 8 that clarified to me that the major errors can be caused in local contributing area estimation causing a subsequent wrong estimation of

hydrograph. To this respect the paper shows that its method brings in quite an improvement in some simulations. Therefore my overall judgment is that the matter is good, but the organization of the paper highly improvable. I suggest to move in evidence the results and let the details to Appendix or to the final part of the paper. To simplify, I suggest to exchange the position of section 4 and section 2, after a preliminary, but brief discussion, of the methods. I understand that this may sound quite a violation of the traditional organization of papers, but I feel that probably in this way the paper should be more appealing to the reader. The epitome if it is the Caption of Figure 1. The phrase "for each cell the representative river pixel (dark red square) inside the effective area (shaded area) and subsequently the outlet pixel (orange square) is identified, and 1B) based on the fine-resolution flow path downstream of the outlet pixel (black lines)," should become: "for each cell, based on the fine-resolution flow path downstream of the outlet pixel (black lines), the representative river pixel (dark red square) inside the effective area (shaded area) and subsequently the outlet pixel (orange square) is identified". The latter form of the phrase makes me understand what they are doing, not the reverse. The explanation in the text, indeed is much more clear than the caption of the Figure, but because I tried to understand first the procedure from the Figure, I had to read more times all of it to understand what was the point. For what regards the errors they find in flood forecasting, while I find interesting the idea that the coarse graining of the river network topology should be made to preserve the main characteristics of the flood wave, It is not clear to me how the errors (except maybe for those connected with the estimation of the contributing areas which bring with them a proportional error in rainfall inputs) were not corrected, at least partially, by a calibration. In my experience, the standard procedure for any hydrological modelling run follows the procedure of calibrating the model parameters and then validating them. The process of calibration, I am sure, introducing effective quantities for instance in those parameters related to flow velocities, can correct delays or anticipations of the flood wave due to a wrong estimation of stream length. Differently would be a problem, since the fractality of rivers. To say in another way, their argument to favor their method of river network representation, could be not so relevant at the end, for modelling. A last note regards the subgrid representation of the networks. I found the paper by V. Casulli (2019) of real interest for the issue.

In summary I believe the paper can be accepted after a major readjusting of its structure.

Reference

Casulli, Vincenzo. 2019. "Computational Grid, Subgrid, and Pixels." International Journal for Numerical Methods in Fluids 90 (3): 140–55.

---

## Author Comment (AC1) · 14 Apr 2021

Response to **Anonymous Referee #1

The authors describe a method to uspscale high-resolution data, based on a back-tracking approach also used by former work of Yamazaki et al. 2009 or Wu et al. 2011). The authors show methods to assess the quality of different methods.

The paper represents significant progress as it is a further development of the concept of Yamazaki et al:

- The authors can show that there approach yields better results in comparison to other methods

- It is open source

- The authors show that their method can be applied in hydrological models towards free scalable models

- It can be used as the common D8 network

The presentation quality and scientific quality is good. A few points could be discussed in a different way.

We are pleased to read that our manuscript is considered significant progress to the reviewer. We would also like to thank the reviewer for the thorough review and comments, which we believe have led to an improvement in the manuscript. Our response to each comment can be found in the paragraphs below. Most importantly we have clarified some points regarding the river width and DMM30 dataset at various sections of the manuscript, improved a figure in the methods section, added a sensitivity analysis for thresholds used in the IHU method and replaced two figures in results section with one new figure which allows for a more quantitative comparison between upscaling methods.

Main point of criticism is that it does not include a link to the work done in the ISI-MIP (https://www.isimip.org/) project. In this project, quite a number of hydrological models use a defined set of input data as a global 30 arcmin setting. The network used here is the DDM30 (Doell and Lehner 2002). It is questionable, if this database is the best choice (see Zhao et al. 2017 https://doi.org/10.1088/1748-9326/aa7250), but it is used as the defined river network. The paper can run without a direct comparison to DDM or a comparison to 30 arcmin, but the value of this paper (and the numbers of citations) can be improved, if it is compared against: a.) 30 arcmin (maybe instead of 15 arcmin, which is rarely used in hydrological models) b.) DDM (maybe instead DMM, as the DMM is not so often used (cited 25 times and DDM cited 147 times)

We agree with the reviewer that DDM30 is an important dataset used in many global hydrological models and have updated the introduction to reflect this better. We deliberately selected DMM instead of the DDM30 method as benchmark for IHU (proposed method) as to our understanding DMM has been re-used as a method with different source datasets (Thober et al., 2019), while the method used for DDM30 (hereafter DDM30 method) was only used to derive the DDM30 dataset. Furthermore the DDM30 dataset also required extensive manual editing (Döll and Lehner, 2002), which makes it impossible to compare the method in an automated manner. Finally, DMM is arguably a slightly more sophisticated method compared to the DDM30 method. The DDM30 method sets the upscaled flow

direction based on the fine-resolution flow direction of the outlet pixel of each coarse-resolution grid cell, where the fine-resolution outlet pixel is the pixel with the largest upstream area within each coarse-resolution grid cell. The DMM uses the same fine-resolution outlet pixel as the starting point, but the upscaled flow direction is set after tracing the fine-resolution flow direction downstream until it leaves a buffered area around the cell of origin (for details see Appendix A) which is arguably a better estimate for the coarse-resolution flow direction. The choice of resolution is based on the increasingly higher resolution of global hydrological models (Bierkens, 2015) as stated in the introduction. Based on this observation most models will likely run on finer spatial resolutions than 30 arcmin which is why the multi-resolution MERIT hydro IHU dataset contains maps up to 15 arcmin resolution.

Using a power function of the upstream area for river width is a weak point here. This does not work for a global dataset and not even for the River Rhine with high runoff in the mountains and low runoff in the lowlands. This approach is not state of the art. The paper says it will provide a parametrization for distr. hydrological models. The approach for providing river width is not appropriate. For sure it would be fine to have the full package incl. river width and Manning's roughness. I think it is still a fine paper, if you exclude river width (as you exclude Manning's anyway)

We agree with the reviewer that this is a simplistic approach which will not yield satisfying results when applied over larger areas with different climate. In our paper we only applied it to the case study experiment to fill data gaps in the existing river width layer, for which we argue that this simplification won't affect the conclusions based on the experiment as it is set up as a sensitivity analysis rather than a validation with observed data. The global river width layer of the MERIT hydro IHU dataset represents river width estimates from the original 3 arcsec MERIT hydro dataset at the fine-resolution outlet pixels of each coarse-resolution grid cell of the dataset and has no data where the underlying data is missing at that point. We have clarified this in the methods and focus on flow direction, sub-grid river length and slope parameters in the abstract and introduction. In the discussion we also mention that the power-law is a strong simplification and refer to better alternatives if applied for a real instead of synthetic event simulations. We would like to keep the river width layer with data gaps as part of the dataset however, as we think it can be a useful starting point to derive a full coverage river width dataset for a region of interest using more advanced methods to fill these gaps, e.g. based on regression techniques similar to Barbarossa et al. (2018).

**Detailed comments:**

60: As I said, I am not a fan of the DDM30, but it is THE reference river network in a global hydrological intercomparison project.

To reflect this we have added a sentence that DDM30 is "used by most global hydrological models within the Inter-Sectoral Impact Model Intercomparison Project"

66: Wu et al. 2011,2012 and Yamazaki (2009) already give out length, Yamazaki already give out slope or elevation at the outlet point.

Thanks for pointing this out. We rephrased to "Furthermore, none of the D8 upscaling methods derive both sub-grid river parameters of length and slope, which are required for many hydrological models."

85 .. often defined by … Isn't it a requirement to be a multiple of the finer grid?

You are correct, we rephrased the sentence to: "IHU requires a target resolution (grey dashed grid lines) which is multiple of the input resolution and two input maps: a fine-resolution flow direction and upstream area map [...]"

115: the equation needs $|x-x0|^{0.5}$ instead of brackets

We have updated equation 1 accordingly.

118: if no output pixel is found… where does this happen in fig 1

We have added a link to the figure to the explanation of step 1-2: "see the trace downstream of the outlet pixel of cell b3 to cell c3 in the example"

126: Maybe a description like in chess B2 instead h would be easier

Thanks for this suggestion. We have updated figures 1 and A1 and the step-by-step description with your suggestion.

154: You have several thresholds in your method e.g. sqrt(R), min upstream_area = 0.25, length of cell = 0.25. Did you test this setting, did you do a sensitive analysis of these values. Where are they from? Maybe for the Rhine it would be good to show some variation of these thresholds

These assumptions have been tested, but were not included in the manuscript. The sqrt(R) is based on EAM (Yamazaki et al., 2008). The length and upstream area thresholds are found by trial and error. Other studies use similar thresholds for minimum length, but with different values, 0.5 in Yamazaki et al (2009) and 0.6-0.8 in Wu et al. (2011). We found that these values are too large when applied to higher resolution target grids compared to the aforementioned papers. We have added appendix E with a sensitivity analysis based on the Rhine basin as suggested by the reviewer and added the following text to section 2.1: "The sensitivity of the *R* parameter to define the effective area in step 1-1 as well as the minimum length and minimum upstream area thresholds used to optimize sub-grid river length in step 3 are tested for the river Rhine basin, see appendix E. As step 2-4 are iterated, the minimum river length and upstream area thresholds may also affect the upscaling accuracy as it may provide room for improvements in the next iteration of step 2. We found that the thresholds change the accuracy of the upscaled maps at less than one percent of the output basin cells see Figure E1. A lower minimum upstream threshold generally has a positive effect on the upscaling accuracy and number of cells with small river length but increases the number of cells with small upstream area. The selected thresholds provide a balance between accuracy and cells with small river length or contributing area, but if the latter is of less importance the minimum upstream area threshold might thus be lowered for improved accuracy."

189: Nice solution of these "orphan" problem – upstream cells that have no direct parents

Thanks.

204-206: This paper does not show a valid way to derive river width for all cells (and it is stated well, you need the parameter for all cells). Filling up with a power function of upstream area will not work. Maybe using some regression/machine learning technique like in Barbarossa et al. 2018 (https://www.nature.com/articles/sdata201852) will help. Maybe dismiss 204-206 and 2F (and write a second paper on width and Manning's). For the routing example, your assumption of width and Manning's is ok

While we agree with the reviewer that a full coverage river width dataset is required for model application, we think the MERIT Hydro IHU river width data layer is still of added value. It provides a useful starting point to derive a full coverage river width dataset for a region of interest. As this is not the focus of the paper we have not tried to derive complete coverage of this data. We have amended the text to reflect this: "The river width is based on the MERIT Hydro width data layer at the outlet pixel. Note that this data contains gaps, i.e. not all outlet pixels have a river width in the underlying data, see Figure 2F which shows river widths in green colors. For global coverage and application in hydrological models these gaps need to be filled which is outside the scope of this paper. This data layer could still function as a starting point for any gap filling based on regression techniques (Andreadis et al., 2013; Barbarossa et al., 2018)"

205: Which outlet pixel in @F does not have a river width in fig 2F. Using river width interpolation with a power function and upstream area is a really weak assumption.

We agree that upstream area is a poor predictor for river width when applied on a large scale. Note that we only use this for a specific experiment in a case study, for which the reviewer agrees it is an OK choice. We have added a sub sentence to explain the figure depicting the river width in 2F better, see previous response.

222: As before: it is a dataset, but an incomplete dataset for kinematic routing missing an adequate solution for river width and channel roughness. I am not asking for these 2 additional parameters, but mentioning that these 2 are missing for a complete dataset (and maybe river depth, too).

We do not want to claim that the dataset is complete, to clarify this we have added the following to the discussion "Besides flow directions, IHU derives additional layers of sub-grid drainage area, river length and slope data, a river width estimates for large rivers and hydrologically adjusted elevation. While these layers cover most parameters required in the routing modules of many hydrological models, for a complete river parameter dataset a full-coverage river width layer is required as well as river bed roughness. For more advanced routing models a river bed level and river bank-full depth might also be required (Yamazaki et al., 2011)."

229: Also mentioned before: DMM is rather special, a comparison to DDM30 would be better

While we agree the DDM30 maps have more applications, the method behind DMM is more re-used and arguably more advanced as explained in an earlier response. Furthermore, a large-scale comparison against the DDM30 method is impossible because of the manual editing applied to the DDM30 map (Döll and Lehner, 2002).

260: Not so clear, why the minimum upstream area is chosen to be 10 km2. Is this done only for the original 3arcsec, or also for the 30 arcsec version? A 1km2 threshold would be more reasonable? Why having a network to 30 arcsec and then aggregating again to 10km2? Adding a reason for 10km2 would be ok.

The runoff in each headwater cell is assumed to drain instantaneously to the river channel. By using this threshold, the area for which we assume instantaneous drainage is more comparable between resolutions. This also follows the implementation in hydrological models such as wflow (Imhoff et al., 2020) where a minimum upstream area is used to define river cells for which river routing is performed. We have clarified this in the text: "By using this threshold, the area of headwater catchments for which we assume instantaneous drainage is more comparable between resolutions."

265-273: For the synthetic runoff event, a simple assumption is ok, like equation 5 and roughness=0.03. What about the river depth to get the perimeter you need for the kinematic routing?

The wetted perimeter is calculated each time step based on the water depth. Basically, this assumes a rectangular profile without floodplains. While we know this is a strong simplification and floodplain storage is a significant process in river routing (Zhao et al., 2017), we argue that for the experiment this is less important as it is set up as a sensitivity analysis rather than a validation with observed data to assess the effect of errors in the flow direction on simulated streamflow.

273: later in357 you describe the synthetic runoff event, It would be better to describe here in more detail how your synthetic event looks like. From your sentence in line 356 I assume: Uniform runoff for the whole Rhine of around 0.2 mm per day (0.002 per 15 min) to reach 500 m/s at run outlet and then increasing to 0.014 mm per 15 min to reach peak of 3000 m/s?

Great suggestion. We have slightly modified the experiment to represent more realistic average and extreme discharge conditions for the Rhine river and added the following text to section 3.2 "The runoff event is triangular shaped with a total duration of 10 days, it starts with 1.2 mm day$^{-1}$ and increases linearly to 6 mm day$^{-1}$ in 5 days after which it decreases back to 1.2 mm day$^{-1}$ in the next 5 days. This yields an initial flow of around 2,700 m$^3$s$^{-1}$ and a peak discharge of around 10,800 m$^3$s$^{-1}$, which roughly corresponds to average and around 1-in-35 year discharge conditions for the Rhine basin at Lobith (Hegnauer et al., 2014)."

350: A table for the Rhine as the table 2 would be good. Maybe even later and including the synthetic runoff results.

See response to comment on line 376.

371: grey line as cumulated runoff?

This should indeed be accumulated runoff, we have changed this in the caption of the figure.

376-404: This part is interesting, but would benefit if it concludes in a method which can be used to compare different methods in numbers e.g. creating a table with flood peak magnitude (btw a flood at around 3000 m3/s is not a flood

in the upper Rhine) and timing like the numbers in line 383 and line 385. But due to the different N, the numbers are not really comparable. How about selecting only locations which all methods have in common, describing a method to find these locations and then it is possible to set up a table with peak magnitude smaller or bigger than 2%, 5%, 10% and flood peak timing different by percent of runoff peak time to routing peak time

We have replaced Figure 8 and 9 with a new summarizing Figure 8 including CDFs which show the difference in flood peak timing and magnitude at different percentiles of the locations, which we think reflects the full distribution of the data better than a table but still allows the reader to easily compare the statistics at different levels. The original Figure 8 and 9 have been moved to the appendix.

384: maybe not an absolute hour, but a percentage of the difference between running time between runoff peak (gray line) and reference time (3 arcsec model). Because it does not matter so much if the delay in a big basin is 2 hours, but it matters if it is in a small basin.

We do agree with the reviewer that the difference in timing should be seen in relation to the upstream area at each location. We have therefore scaled the markers in Figure 8 with the upstream area. We could not think of a method to express the difference in timing as a percentage as it is not clear what the reference should be.

442 As before: river width I would delete from this list as you cannot derive it in a proper way with IHU.

We have amended this part in the discussion; "To achieve complete coverage of river width, the sub-grid river width data requires to be interpolated for data gaps and lake and reservoir areas if these were to be modelled explicitly in the routing model. Here, we used a strongly simplified power-law relationship between width and upstream area. For applications for real events instead of sensitivity analysis with synthetic data, this estimate should be improved using the well-established geomorphic relationships between bank-full discharge and river depth as proposed in the downstream hydraulic geometry framework (Leopold and Maddock, 1953; Savenije, 2003), for instance by the clustering approach proposed by Andreadis et al. (2013) or additional river width data from e.g. Allen and Pavelsky (2018)"

448-444: To my understanding, there is no well-established geomorphic relationship between discharge and river depth nor river width. It is mostly regression between discharge, upstream area, etc. like in Pistocci 2006 or others. A way to improve this is machine learning and feeding it with climatic, geomorphic data like Barbarossa et al. 2018 or using advanced technics of remote sensing e.g., Allen and Pavelsky 2018 or a combination of both. Your approach for having a full dataset for benchmarking is ok, but maybe dismiss the part for river width.

There is a geomorphological relation between width and discharge. According to Lacey's formula the width of a natural channel at bankfull capacity is proportional to the root of the discharge, this is well explained by Savenije et al (2003). This reference is added to the discussion, see previous response.

465: The Merit DEM (Yamazaki et al. 2017) is a very good DEM. But mostly due to anthropogenic overprinting, rivers do not always follow the lowest elevation. They are redirected or even on a higher level than the surrounding

landscape (there are many examples of this in the Netherlands). One way to improve the DEM would be to burn in the existing river network. Not part of your work here, but maybe something to think of in the discussion.

In the case of this paper we have used the MERIT Hydro DEM and flow direction dataset (Yamazaki et al., 2019) which is derived by burning vector files of channels and rivers from various sources in the MERIT DEM. Of course these data also have limitations as pointed out by the reviewer. For this paper however, we focus on flow direction upscaling and assume that the flow directions of the original high-resolution data are correct. We have added this assumption to the discussion: "In this study we benchmarked several upscaling methods based on the same baseline hydrography data. This choice was made to focus the paper on differences in upscaling methods, where we assume the underlying high-resolution data is correct. For future studies it would be interesting to also compare [...]"

**References:**

Andreadis, K. M., Schumann, G. J.-P. and Pavelsky, T. M.: A simple global river bankfull width and depth database, Water Resour. Res., 49(10), 7164–7168, doi:10.1002/wrcr.20440, 2013.

Barbarossa, V., Huijbregts, M. A. J., Beusen, A. H. W., Beck, H. E., King, H. and Schipper, A. M.: FLO1K, global maps of mean, maximum and minimum annual streamflow at 1 km resolution from 1960 through 2015, Sci Data, 5(1), 180052, doi:10.1038/sdata.2018.52, 2018.

Bierkens, M. F. P.: Global hydrology 2015: State, trends, and directions, Water Resour. Res., 51(7), 4923–4947, doi:10.1002/2015wr017173, 2015.

Döll, P. and Lehner, B.: Validation of a new global 30-min drainage direction map, J. Hydrol., 258(1–4), 214–231, doi:10.1016/S0022-1694(01)00565-0, 2002.

Savenije, H. H. G.: The width of a bankfull channel; Lacey's formula explained, J. Hydrol., 276(1–4), 176–183, doi:10.1016/S0022-1694(03)00069-6, 2003.

Thober, S., Cuntz, M., Kelbling, M., Kumar, R., Mai, J. and Samaniego, L.: The multiscale routing model mRM v1.0: simple river routing at resolutions from 1 to 50 km, Geoscientific Model Development, 12(6), 2501–2521, doi:10.5194/gmd-12-2501-2019, 2019.

Wu, H., Kimball, J. S., Mantua, N. and Stanford, J.: Automated upscaling of river networks for macroscale hydrological modeling, Water Resour. Res., 47(3), doi:10.1029/2009WR008871, 2011.

Yamazaki, D., Masutomi, Y., Oki, T. and Kanae, S.: An Improved Upscaling Method to Construct a Global River Map, in Proceedings of the 4th Asia-Pacific Hydrology and Water Resources (APHW) Conference, Beijing., 2008.

Yamazaki, D., Oki, T. and Kanae, S.: Deriving a global river network map and its sub-grid topographic characteristics from a fine-resolution flow direction map, Hydrol. Earth Syst. Sci., 13(11), 2241–2251, doi:10.5194/hess-13-2241-2009, 2009.

Yamazaki, D., Kanae, S., Kim, H. and Oki, T.: A physically based description of floodplain inundation dynamics in a global river routing model, Water Resour. Res., 47(4), 1–21, doi:10.1029/2010WR009726, 2011.

Yamazaki, D., Ikeshima, D., Sosa, J., Bates, P. D., Allen, G. H. and Pavelsky, T. M.: MERIT hydro: A high-resolution global hydrography map based on latest topography dataset, Water Resour. Res., 55(6), 5053–5073, doi:10.1029/2019wr024873, 2019.

Zhao, F., Veldkamp, T. I. E., Frieler, K., Schewe, J., Ostberg, S., Willner, S., Schauberger, B., Gosling, S. N., Schmied, H. M., Portmann, F. T., Leng, G., Huang, M., Liu, X., Tang, Q., Hanasaki, N., Biemans, H., Gerten, D., Satoh, Y., Pokhrel, Y., Stacke, T., Ciais, P., Chang, J., Ducharne, A., Guimberteau, M., Wada, Y., Kim, H. and Yamazaki, D.: The critical role of the routing scheme in simulating peak river discharge in global hydrological models, Environ. Res. Lett., 12(7), 075003, doi:10.1088/1748-9326/aa7250, 2017.

---

## Author Comment (AC2) · 14 Apr 2021

Response to **Anonymous Referee #2

I am sorry I took so long for reviewing this paper. This is not due to the topic the paper deals with but the way the paper is organised. I do not want to appear definitive, however, if the topic, of adapting the representation of a river network at certain resolution to be the more effective as possible, is of geographical interest, I am still not convinced it is also of hydrological importance. The Authors should work a little more to make the paper more appealing, instead they buried the good ideas under a lot of technicalities which are, certainly, important but secondary to the reasons why they do it. Finally some light came in with Figure 7 and 8 that clarified to me that the major errors can be caused in local contributing area estimation causing a subsequent wrong estimation of hydrograph. To this respect the paper shows that its method brings in quite an improvement in some simulations. Therefore my overall judgment is that the matter is good, but the organization of the paper highly improvable.

We appreciate the time you took to review our manuscript and are happy to read that you find the content good and agree the method has benefits for hydrological simulations. We will address your points of concern in more detail in the following responses. Most importantly, we have tried to clarify some points in the introduction and tried to make the structure of the paper more appealing.

I suggest to move in evidence the results and let the details to Appendix or to the final part of the paper. To simplify, I suggest to exchange the position of section 4 and section 2, after a preliminary, but brief discussion, of the methods. I understand that this may sound quite a violation of the traditional organization of papers, but I feel that probably in this way the paper should be more appealing to the reader.

We agree the paper is rather technical, but would like to argue that the purpose of the paper was to present and demonstrate a new method, which falls in scope with HESS. We disagree that our new method is not of "hydrological importance". We have demonstrated that our method significantly and positively influences the quality of direction of streams within a network, and as a consequence, also simulated peak discharge timing and magnitude. We have made some changes to the manuscript to clarify this and to make the structure more appealing:

- We have clarified the purpose of the paper as well as the relevance of good flow direction data and flow direction upscaling methods better in the introduction. As explained above, the purpose is firstly the presentation and secondly the demonstration for river routing of the new method.
- This is largely supported by the chosen structure, which is why we have decided not to change the structure significantly, but to resent the technicalities of the new method in a new section "2. Iterative Hydrography Upscaling" before section "3. Methods" which deals with the benchmark and synthetic runoff experiment
- We have added a more high-level introduction to "2. Iterative Hydrography Upscaling" to provide the reader with some more guidance before diving into the technicalities.

The epitome if it is the Caption of Figure 1. The phrase "for each cell the representative river pixel (dark red square) inside the effective area (shaded area) and subsequently the outlet pixel (orange square) is identified, and 1B) based

on the fine-resolution flow path downstream of the outlet pixel (black lines)," should become: "for each cell, based on the fine-resolution flow path downstream of the outlet pixel (black lines), the representative river pixel (dark red square) inside the effective area (shaded area) and subsequently the outlet pixel (orange square) is identified". The latter form of the phrase makes me understand what they are doing, not the reverse. The explanation in the text, indeed is much more clear than the caption of the Figure, but because I tried to understand first the procedure from the Figure, I had to read more times all of it to understand what was the point.

While we have put a lot of effort in creating an explaining figure, it remains difficult to summarize the entire method in a caption. We understand this sentence is unclear and have modified it to read: "Firstly, 1A) the representative river pixel (dark red square) inside the effective area (shaded area) and the outlet pixel (orange square) are identified for each cell based on upstream area, then 1B) the fine-resolution flow path downstream of the outlet pixel (black lines) is traced to a neighboring cell, 1C) to set the initial flow direction (orange arrow)"

For what regards the errors they find in flood forecasting, while I find interesting the idea that the coarse graining of the river network topology should be made to preserve the main characteristics of the flood wave, It is not clear to me how the errors (except maybe for those connected with the estimation of the contributing areas which bring with them a proportional error in rainfall inputs) were not corrected, at least partially, by a calibration. In my experience, the standard procedure for any hydrological modelling run follows the procedure of calibrating the model parameters and then validating them. The process of calibration, I am sure, introducing effective quantities for instance in those parameters related to flow velocities, can correct delays or anticipations of the flood wave due to a wrong estimation of stream length. Differently would be a problem, since the fractality of rivers. To say in another way, their argument to favor their method of river network representation, could be not so relevant at the end, for modelling. OVERFITTING physical realistical.

The aspect of model calibration can indeed resolve some errors related to incorrect representation of river parameters. Errors in upstream river length can potentially be compensated by manning roughness values using calibration. Other local errors in the flow direction data, such as a confluence which is located upstream from the actual position, will however lead to large local errors which cannot be compensated by calibrating other river parameters as the distribution of the runoff is simply incorrect. But more important, using calibration to compensate for erroneous data in your model will likely lead to improved simulations but for the wrong reasons (it might for instance take physically unrealistic manning parameters to compensate for errors in river length) and will likely lead to overfitting of the model which might decrease its performance outside the range of the calibration data (Hrachowitz et al., 2013; Kirchner, 2006).

A last note regards the subgrid representation of the networks. I found the paper by V. Casulli (2019) of real interest for the issue.

Thanks for this interesting reference.

In summary I believe the paper can be accepted after a major readjusting of its structure.

Thank you - we have adjusted the structure as summarised in response to earlier comment.

**References:**

Hrachowitz, M., Savenije, H. H. G., Blöschl, G., McDonnell, J. J., Sivapalan, M., Pomeroy, J. W., Arheimer, B., Blume, T., Clark, M. P., Ehret, U., Fenicia, F., Freer, J. E., Gelfan, A., Gupta, H. V., Hughes, D. A., Hut, R. W., Montanari, A., Pande, S., Tetzlaff, D., Troch, P. A., Uhlenbrook, S., Wagener, T., Winsemius, H. C., Woods, R. A., Zehe, E. and Cudennec, C.: A decade of Predictions in Ungauged Basins (PUB)—a review, Hydrol. Sci. J., 58(6), 1198–1255, doi:10.1080/02626667.2013.803183, 2013.

Kirchner, J. W.: Getting the right answers for the right reasons: Linking measurements, analyses, and models to advance the science of hydrology, Water Resour. Res., 42(3), doi:10.1029/2005wr004362, 2006.

---

## Author Response (AR2)

Reviewer #1

A hydrography upscaling method for scale invariant parametrization of distributed hydrological models

I agree with the authors that the paper has significantly improved.

My questions from the former review have been answered in a very good way.

The paper is addressing a relevant problem in hydrological modeling and presents an improved solution to the existing methods.

Furthermore the solution is available as open source python code.

I tested the code. It is working and it is a fine piece of open source software which include code, license information, documentation, testing, coverage, changelog.

Even if this is not part of the paper, I put my ranking of the scientific quality to excellent also because of this round software package.

I still think it would have been a benefit to test the method against the DDM30 network at 30arcmin, because this is used at the moment as reference in the global hydrological model intercomparison project. But I accept that this paper uses 2 other methods to test against on 0.5, 5 and 15 arcmin.

Thanks for your review and constructive feedback in both review rounds!

Small remarks:

100: While it might be possible to use your routine on the fly, meaning you have a high resolution dataset and convert all data to a lower resolution, it might be more convenient to do a preprocessing and store the low resolution before you run the hydrological model. This will save memory and time. And also your tool needs to keep at least part of high resolution of d8 in memory. I cannot see this as a real disadvantage (a real disadvantage is the DRT is not open source, and one is limited to certain resolutions and a WGS84 lat/lon projections – maybe this changed meanwhile)

Line 69 now reads: "While DTR has proven successful at automatically upscaling 30 arcsec flow direction data to coarser resolutions (Wu et al., 2012), it has not been applied to higher resolution data to the best of our knowledge and its code not open source available.

270: Not clear if or where there are data gaps in figure 2F

Line 229 now reads: "Note that this data contains gaps, see **Error! Reference source not found.**F, which shows that not all outlet pixels (grey squares) have a river width (green colors) in the underlying data. "

498/535ff It seem your approach show much better results in cases where rivers run parallel for some time (e.g. Rhine and Meuse). You point this out in 535ff but it seems your approach is giving better results for parallel rivers even with the D8 limitation. Maybe point this out that your approach is superior to the others.

Line 462 now reads: "Although much less compared to EAM and DMM, erroneous IHU upscaled flow directions are still found in [...]"